# Modification of RNF183 via m6A Methylation Mediates Podocyte Dysfunction in Diabetic Nephropathy by Regulating PKM2 Ubiquitination and Degradation

**DOI:** 10.3390/cells14050365

**Published:** 2025-03-01

**Authors:** Dongwei Guo, Yingxue Pang, Wenjie Wang, Yueying Feng, Luxuan Wang, Yuanyuan Sun, Jun Hao, Fan Li, Song Zhao

**Affiliations:** 1Department of Pathology, Hebei Medical University, Shijiazhuang 050017, China; 13844646919@163.com (D.G.); pangyx2001@163.com (Y.P.); 15603523344@163.com (W.W.); 17713277211@163.com (Y.F.); 18633014085@163.com (L.W.); 17320695832@163.com (Y.S.); haojun@hebmu.edu.cn (J.H.); 2Hebei Key Laboratory of Kidney Diseases, Shijiazhuang 050017, China; 3Center of Metabolic Diseases and Cancer Research, Institute of Medical and Health Science of Hebei Medical University, Shijiazhuang 050017, China; 4Hebei Provincial Key Laboratory of Medical Imaging Science, Shijiazhuang 050017, China

**Keywords:** diabetic kidney disease, RNF183, m6A methylation, ubiquitination, PKM2

## Abstract

Diabetic kidney disease (DKD) is a prevalent complication associated with diabetes in which podocyte dysfunction significantly contributes to the development and progression of the condition. Ring finger protein 183 (RNF183) is an ER-localized, transmembrane ring finger protein with classical E3 ligase activity. However, whether RNF183 is involved in glomerular podocyte dysfunction, which is the mechanism of action of DKD, is still poorly understood. In this study, we first demonstrated that RNF183 expression in glomerular podocytes of patients with DKD decreased as the disease progressed. Additionally, our transcriptome sequencing analysis of kidney tissues from diabetic mice revealed a significant reduction in RNF183 expression within the kidney cortex. Similarly, the expression of RNF183 was significantly reduced both in the kidneys of diabetic mice and in human podocytes exposed to high glucose conditions. The downregulation of RNF183 resulted in a suppression of autophagic activity, an increase in apoptotic cell death, and reduced expression of cellular markers in HPC cells. We found that RNF183 was modified via N6-methyladenosine (m6A) RNA methylation. Meanwhile, treatment with meclofenamic acid 2 (MA2), an m6A demethylase inhibitor, resulted in the upregulation of RNF183 expression in HPC cells cultured in high glucose conditions. Furthermore, high glucose treatment decreased the transcription and protein levels in both the m6A writer methyltransferaselike3 (METTL3) and the m6A reader insulin-like growth factor 2 mRNA-binding protein 2 (IGF2BP2). IGF2BP2 assisted with METTL3, which is jointly involved in the transcription of RNF183. Furthermore, we confirmed that RNF183 directly ubiquitinates M2 pyruvate kinase (PKM2) through co-immunoprecipitation (Co-IP) and liquid chromatography–mass spectrometry (LC-MS) experiments. The level of PKM2 ubiquitination was increased following RNF183 overexpression, leading to enhanced PKM2 protein degradation and subsequently alleviating high glucose-induced podocyte damage. The results of this study indicated that RNF183 was regulated via m6A methylation modification and that RNF183 expression was reduced in HPC cells treated with high glucose, which resulted in decreased PKM2 ubiquitination levels and subsequently aggravated podocyte injury. The findings suggest that RNF183 may serve as a potential therapeutic target for diabetic kidney injury, offering new insights into its role in the progression of DKD.

## 1. Introduction

Diabetes mellitus (DM) is a common chronic metabolic disease. As the global population grows and ages and obesity increases, the incidence of diabetes is increasing annually, making it one of the major diseases threatening human health [1,2,3]. The characteristics of Type 1 diabetes (T1DM) are the destruction of pancreatic β cells and insulin deficiency, while Type 2 diabetes (T2DM) is primarily caused by insulin resistance (IR) and insufficient insulin secretion. Type 2 diabetes can cause multi-organ damage and various complications [4]. DKD is the most common complication of diabetes. More than 40% of diabetic patients have renal lesions, which result in persistent proteinuria and progressive renal dysfunction, ultimately leading to end-stage renal disease (ESRD), which is a condition that can be life-threatening [5,6]. Currently, hypoglycemic drugs and intensive glycemic control are not effective at improving kidney damage in diabetic patients. Due to the lack of effective targets, the treatment of DKD is limited to symptomatic treatment, such as the application of renin–angiotensin system blocking agents, sodium-dependent glucose transporter 2 (SGLT-2) inhibitors, and semaglutide to reduce proteinuria [7]. Therefore, it is very important to explore the pathogenesis of DKD to identify potential targets and reasonable interventions, which are crucial for the prevention and treatment of this disease.

Podocytes are highly specialized and terminally differentiated cells that are attached to the lateral basement membrane of the glomerulus. Podocytes contribute to the formation of the glomerular filtration barrier, support the maintenance of normal kidney structure and function, and play a central role in the pathological processes underlying proteinuric nephropathy. Increasing evidence indicates that abnormal podocyte function is an important event triggering DKD [5,8]. In the diabetic environment, podocytes themselves exhibit various dysfunctions, such as reduced levels of diaphragm-associated proteins in the podocyte processes, cytoskeletal protein rearrangement, autophagy inhibition, and mitochondrial pathway apoptosis.

Ring finger protein 183 (RNF183), which maps to the endoplasmic reticulum (ER), belongs to the E3 ligase family. This protein exhibits E3 ubiquitin ligase activity and plays an essential role in the ubiquitination process [9]. Recent studies have shown that RNF183 in HeLa cells mediates ER stress and ER-induced apoptosis through ubiquitinated Bcl-xL [10]. The expression of RNF183 was also found to be significantly lower in high glucose-cultured HPC cells than in normal glucose-treated podocytes. However, to date, the contribution of RNF183 to DKD development remains unexplored in the existing literature. This study explores the pathogenesis of DKD by targeting RNF183.

We further explored the mechanism of the downregulation of RNF183 expression in diabetic podocytes, and m6A is the most extensive RNA modification in eukaryotes, which has received wide interest in recent years. Usually, the m6A modification occurs in the 3′-UTR region near the translation stop codon and is embedded in a conserved sequence 5′-RRACU-3 to achieve regulation of RNA by binding to a specific methylated reading protein [11]. The m6A methylation modification consists of three parts consisting of the m6A author, m6A erasure, and m6A reader. m6A is written by m6A methyltransferases, including methyltransferaselike3 (METTL3), methyltransferaselike14 (METTL14), and Wilms tumor 1-associated protein (WTAP). The removal of m6A demethylase, including AlkB homology 5, RNA demethylase (ALKBH5), and fat mass and obesity-related protein (FTO) [12], is recognized by m6A binding proteins, including YT521-B homology (YTH) domain-containing proteins (YTHDCs, including YTHDC1/2), YTH domain family proteins (YTHDFs, including YTHDF1/2/3), insulin-like growth factor 2 mRNA-binding proteins (IGF2BPs, including IGF2BP1/2/3), and eukaryotic translation initiation factor 3 (eIF3) [13,14]. Notably, it was previously reported that circ-0,000,953 expression was significantly reduced in glomerular podocytes from diabetic mice. METTL3 was found to regulate the expression and methylation of circ-0,000,953 through the interaction with YTH N6-methyladenosine RNA binding protein 2 (YTHDF2). In the context of DKD, circ-0,000,953 played a critical role in modulating podocyte autophagy by targeting the Mir665-3p-Atg4b signaling axis [15]. In bladder cancer patients, Yan, B. et al. observed that GLUT3 influences the stability of YTHDC1 by promoting the expression of RNF183, thereby playing a pivotal role in the regulation of both glucose metabolism and the progression of bladder cancer [16]. In our study, we found that IGF2BP2 assisted METTL3 and was involved in the transcription of RNF183.

This experimental study showed that the downregulation of RNF183 was significantly reversed using drug treatment of the demethylase inhibitor MA2 in HPC cells cultured with high glucose. The expression of RNF183 was significantly downregulated after transfection of normal cultured HPC cells with the methylase METTL3 knockdown plasmid. Similarly, RNF183 expression was also significantly downregulated after transfection of normal cultured HPC cells with the IGF2BP2 knockdown plasmid, which may be related to the reduced levels of intracellular RNF183 m6A modification.

M2 pyruvate kinase (PKM2) is a rate-limiting enzyme of glycolysis, which is widely considered and is an important part of regulating cell metabolism and energy metabolism [17]. PKM2 was upregulated in HPC cells from renal biopsies and high glucose cultures from patients with DKD [18]. Moreover, molecular mechanism studies revealed that RNF183 can bind to PKM2 and induce the degradation of the PKM2 protein via the ubiquitination pathway. Thus, reduced RNF183 expression in the diabetic setting accumulates PKM2, ultimately leading to the downregulation of apoptosis, autophagy-related proteins, and cytoskeletal proteins in HPC cells. Therefore, RNF183 may lead to HPC cell dysfunction and the development of DKD through PKM2.

In conclusion, we propose the hypothesis that in HPC cells stimulated with high glucose, IGF2BP2 assists METTL3, reduces the m6A modification level of RNF183, and prevents PKM2 degradation via the RNF183 ubiquitination pathway by aggravating HPC cell dysfunction and the occurrence of diabetic nephropathy. This study aimed to elucidate the role of RNF183 in the pathogenesis of DKD, discover the targets of RNF183, and explore the underlying mechanisms.

## 2. Materials and Methods

### 2.1. Materials

Urine creatinine (C011-2-1) and urinary albumin (A028-2-1) test kits were purchased from Nanjing Jiancheng Bioengineering Institute, Nanjing, Jiangsu, China. RNF183 (ARP43404) was purchased from Shanghai Baili Biotechnology Co., Ltd., Shanghai, China. Primary antibodies against nestin (sc-23927) were purchased from Santa Cruz Biotechnology Co., Ltd., Santa Cruz, Bolivia. Primary antibodies against nestin (ab6142), synaptopodin (ab259976), Nephrin (ab58968), and m6A (ab286164) were purchased from Abcam company, Cambridge, MA, USA. Primary antibodies targeting P62 (18420-1-AP), LC3 (14600-1-AP), BCL-2 (12789-1-AP), BAX (50599-2-AP), ALKBH5 (16837-1-AP), IGF2BP1 (22803-1-AP), IGF2BP2 (11601-1-AP), PKM2 (15822-1-AP), RACK1 (27592-1-AP), Flag (66008-2-Ig), and IgG (30000-0-AP) were purchased from Proteintech Group, Inc., Rosemont, IL, USA. The primary antibodies against METTL3 (96391T), METTL14 (51104T), FTO (45980S), and WTAP (56501T) were purchased from Cell Signaling Technology (CST) Co., Danvers, MA, USA. Primary antibodies against calreticulin (ET1608-60), IgG (HA1027), and IGF2BP3 (EM1706-1-AP) were purchased from Hua’an Biotechnology Co., Ltd., Hangzhou, Zhejiang, China. The primary antibody β-actin (AC026) and HighGene transfection reagents were obtained from Wuhan Aibotek Biotechnology Co., Ltd., Wuhan, Hubei, China. BamH1 (1010S), HindIII (1060S), SalI (1080S), and the PrimeScript™ RT Reagent Kit with gDNA Eraser and SYBR Premix Ex Taq™ II (Tli RNaseH Plus) were obtained from Takara Co., Otsu, Shiga, Japan. The EndoFree Maxi Plasmid Kit (DP117) was purchased from Tiangen Biochemical Technology Limited Technology, Beijing, China. Radio-immunoprecipitation assay (RIPA) lysis buffer (P0013B) reagent was purchased from Biyuntian Biotechnology Co., Ltd., Shanghai, China. The anti-Flag M2 affinity gel was obtained from Merck Medical Technologies in Darmstadt, Germany. The immunohistochemistry standard procedure (SP) kit (SP9000) was obtained from Zhongshan Golden Bridge Technology Co., Beijing, China. Secondary antibodies labeled with DyLight 405, 488, and 594 goat IgG were sourced from KPL, Gaithersburg, MD, USA. The plasmids pGenesil-1-METTL3 and pGenesil-1-IGF2BP2 were retained plasmids in our laboratory. The PCMV3-RNF183-GFP, PCMV3-RNF183-Flag, PCMV3-Flag-IGF2BP2, and PCMV3-PKM2-Flag expression plasmids were obtained from Shenzhou Technology Co., Ltd., Beijing, China. RNF183 mutant plasmids (pCI-neo-GFP-N-RNF183-WT, pCI-neo-GFP-N-RNF183-TM, pCI-neo-GFP-N-RNF183-CS, and pCI-neo-GFP-N-RNF183-KR) were a gift from Professor Junjie Hu (Institute of Biophysics, Chinese Academy of Sciences, Beijing, China). Cycloheximide (CHX), 10 mmol/L; 5-Aza-2′-deoxycytidine (5-Aza), 10 mmol/L; peptide aldehyde proteasome inhibitor Z-Leu-Leu-Leu-CHO (MG132), 10 mmol/L; and trichostatin A (TSA), 50 mmol/L, were obtained from MCE Co., Monmouth Junction, NJ, USA, and used at a dilution of 1:1000. Meclofenamic acid 2 (MA2), also at 50 mM, was generously provided by Professor Caiguang Yang from the Shanghai Institute of Materia Medica, Chinese Academy of Sciences, and used at a dilution of 1:1000.

### 2.2. Human Renal Specimens

Renal biopsy samples from patients with biopsy-proven DKD were obtained from the Nephrology Department of the Second Hospital of Hebei Medical University, and the human renal puncture specimens we collected were all middle-aged and elderly and all patients with Type 2 diabetes. Non-tumor renal tissue from non-diabetic patients undergoing renal cell carcinoma resection was used as a control from the Department of Pathology at the Second Hospital of Hebei Medical University. Paraffin-embedded sections (3 µm) of human kidney biopsies were prepared and used for immunohistochemical staining. All studies involving human samples were approved by the Clinical Medical Ethics Committee of Hebei Medical University. This study was conducted in accordance with the Declaration of Helsinki and approved by the Scientific Research Ethics Committee of the Second Hospital of Hebei Medical University (approval code: 2023-R376; approval date: 30 June 2023). All participants provided written informed consent.

### 2.3. Animals

Eighteen 8-week-old mice, nine male C57BL/6J *db/db* mice and nine male C57BL/6J *db/m* mice (30–35 g), were purchased from Gem Pharmaceutical Technology Co., Ltd., Nanjing, China. The *db/db* mice exhibit obesity and insulin resistance due to carrying a mutated leptin receptor gene. All animal studies were conducted in accordance with the Guidelines for the Care and Use of Laboratory Animals and were approved by the Institutional Animal Care and Ethics Committee of Hebei Medical University (approval code: IACUC-Hebmu-2021046; approval date: 14 December 2021). All mice were housed in an air-conditioned room at room temperature (22–23 °C) with humidity maintained at 40–60%, under a 12 h light/dark cycle. The mice had free access to food (11001A, Hua Fu Kang Biotechnology Co., Ltd., Beijing, China) and water (autoclaved tap water). After being raised to 16 weeks of age, they were placed in metabolic cages for housing, and 24 h urine samples were collected for experiments. All mice were anesthetized with isoflurane, and experiments were conducted once the mice lost consciousness. After the experiment, the experimenter performed cervical dislocation on the mice. The thumb and index finger were placed on both sides of the base of the skull at the neck of the mouse, and with the other hand, the tail was quickly pulled at the base, resulting in the separation of the cervical vertebrae from the skull for euthanasia. All animal experiments were greater than or equal to six independent experiments.

### 2.4. Plasmid Constructs and Transfection

The pGenesil-1 plasmid was laboratory retained and used to construct the shRNA plasmid. The target sequence of the RNF183 gene (5′-CCACTCTTTGAGGGAGTGTTT-3′) was designed from the website https://www.sigmaaldrich.cn/CN/zh “URL (Visit Date: 21 July 2021)”, and the corresponding single-stranded DNA oligonucleotides for the RNF183 shRNA were chemically synthesized. This oligonucleotide was annealed to form a double-stranded DNA fragment. The pGenesil-1 plasmid was digested with BamHI and HindIII restriction enzymes at 37 °C for 3 h. After gel extraction of the products, T4 ligase was used to ligate the products with the double-stranded DNA fragment at 16 °C overnight. The recombinant plasmid was then transformed into Escherichia coli DH5α strain for amplification, and plasmid DNA was extracted using the TIANGEN kit. Due to the presence of a SalI recognition site in the single-stranded DNA oligonucleotide, a 400 bp fragment was obtained after SalI digestion. The successfully constructed plasmid was named pGenesil-1-RNF183. HighGene transfection reagents were purchased from ABclonal Biotechnology Co., Wuhan, Hubei, China. Transfection of HPC cells was carried out following the instructions of the reagents. The detailed operation steps are as follows: We vortexed the plasmid (4 μg), serum-free RPMI-1640 (200 μL), and transfection reagent (8 μL) and incubated the mixture at room temperature for 15 min. Finally, the compound was added evenly dropwise into the wells of 6-well cell culture plates. After 6 h of transfection, half of the transfection medium was replaced with the specified medium, and the corresponding detection was performed after continued culture for 48 h.

### 2.5. Cell Culture and Groups

Human podocytes (HPC, CE28120) were purchased from Beijing Keruisi Bio-Tech Co., Ltd., Beijing, China, and the cells were maintained in the laboratory. The immortalized HPC cells were achieved by introducing the hTERT gene. hTERT is a gene that encodes the catalytic subunit of the telomerase protein, which is positively correlated with telomerase activity. By viral-mediated expression of hTERT, telomerase in normal cells was activated, leading to the elongation of telomeric DNA and the extension of the cell cycle, thus achieving cell immortalization. HPC cells were cultured in RPMI-1640 at 37 °C with 5% CO_2_, supplemented with 10% fetal bovine serum, 100 U/mL penicillin, and 100 μg/mL streptomycin. HPC cells were adherent cells that grew in a typical (epithelial) cobblestone morphology. The medium was changed every 2–3 days, and cells were passaged every 3-4 days at a ratio of 1:2. To investigate the effect of high glucose on cells, the cells were randomly divided into two groups: one exposed to normal glucose (5.5 mmol/L, N) and the other to high glucose (40 mmol/L, H). To explore the affect high glucose on RNF183 expression. We divided HPC cells into six groups, namely, cells treated with high glucose for 0 h, 6 h, 12 h, 24 h, 36 h and 48 h. We divided HPC cells cultured in normal glucose into two groups: pGenesil-1 group and pGenesil-1-RNF183 group. HPC cells cultured in high glucose were divided into four groups: N, H, H with PCMV3-GFP, and H with PCMV3-GFP-RNF183. To explore the mechanism of downregulation of RNF183 transcription in HPC cells stimulated by high glucose, HPC cells were randomly divided into blank control, dimethyl sulfoxide (DMSO) control, histone deacetylase inhibitor trogutin A (TSA), RNA demethylase inhibitor meclofenamic acid 2 (MA2), and DNA methyltransferase (DNMT) inhibitor 5-aza-2′-deoxycytidine (5-Aza). To investigate the effect of m6A methylation on RNF183 expression, HPC cells were divided into pGenesil-1, pGenesil-1-METTL3, and pGenesil-1-IGF2BP2 groups. To investigate the effect of the E3 ubiquitin ligase RNF183 on the expression of the protein PKM2. Given that RNF183 functions as an E3 ubiquitin ligase, its potential impact on PKM2 was further investigated. HPC cells stimulated with high glucose were divided into PCMV3-Flag, PCMV3-Flag-RNF183, and PCMV3-Flag-PKM2 groups. To investigate the type of RNF183-induced PKM2 ubiquitination, HPC cells transfected with PCMV3-Flag-PKM2 for 48 h and treated with MG132 for 6 h were randomly divided into blank control and pCI-neo-GFP-RNF183 plasmids (WT, ΔTM, CS, and KR) groups. HPC cells were co-transfected with PCMV3-HA-Ub and PCMV3-Flag-PKM2, followed by a 48 h incubation and treatment with MG132 for 6 h. After treatment, the cells were randomly assigned to three groups: blank control and pCI-neo-GFP-RNF183 plasmid (WT, Δ TM, CS, and KR) groups. All cell experiments were greater than or equal to 3 independent experiments.

### 2.6. Real-Time PCR

The RNA-Easy reagent was purchased from Vazyme Corporation, Nanjing, China, and RNA samples were obtained from HPC cells or grinding kidney tissue extracted with RNA-Easy reagent, and RNA concentration was examined. The cDNA was synthesized using a reverse transcription kit. For RT-PCR analysis, the SYBR Green kit was employed to detect the samples. The primers used in the experiment were as follows: human RNF183 sense: 5′-CCCTTCAACAACACGTTCCAT-3′; human RNF183 antisense: 5′-CGTGGGCAAGTCAGTGACAG-3′; human 18 S sense: 5′-AT CCTCAGTGAGTTCTCCCG-3′; human 18 S antisense: 5′-CTTTGCCATCACTGCCATTA-3′; mouse RNF183 sense: 5′-CCCTCACTTCCGGATCTTCG-3′; mouse RNF183 antisense: 5′-CCATGCCCCAAAAGAACTGC-3′; mouse 18 S sense: 5′-ACACGGACAGGATTGACAGA-3′; and mouse 18 S antisense: 5′-TTCTTCAGCCTCTCCAGGTC-3′. The data were analyzed using the 2^−ΔΔCt^ method with gene 18 S as the internal control, and all experiments were repeated independently three times to ensure consistency.

### 2.7. Protein Extraction and Western Blot Analysis

Protein samples were obtained by cracking HPC cells or grinding kidney tissue with RIPA lysis buffer and protein concentrations were detected. Protein samples of different experimental groups were subjected to sodium dodecyl sulfate-polyacrylamide gel electrophoresis (SDS-PAGE). After electrophoresis, the proteins are transferred to a polyvinylidene fluoride (PVDF) film in an ice bath (Millikon, Bedford, MA, USA). The PVDF membrane was blocked with serum for 1.5 h at room temperature. Next, We used a series of primary antibodies, including anti-RNF183 (1:800), anti-nestin (1:1000), anti-synaptopodin (1:1000), anti-Nephrin (1:1000), anti-P62 (1:2000), anti-LC3 (1:1500), anti-BCL-2 (1:1500), anti-BAX (1:15,000), anti-METTL3 (1:1000), anti-METTL14 (1:1000), anti-WTAP (1:1000), anti-ALKBH5 (1:1500), anti-FTO (1:1500), anti-IGF2BP1 (1:1500), anti-IGF2BP2 (1:1000), anti-PKM2 (15,822-1AP), anti-RACK1 (1:1500), Flag (66008-2-Ig), anti-IGF2BP3 (1:1500), and anti-β-actin antibody (1:20,000). The PVDF membrane was incubated with specific primary antibodies for 12 h at 4 °C. Then, the PVDF membrane was incubated with secondary antibody of horseradish peroxidase for 1.5 h at room temperature. Finally, the protein expression was visualized by a chemiluminescent solution derived from Tiangen, Beijing, China. All experiments were independently repeated three times. For quantification, the intensity of the strips was measured using ImageJ 1.8.0 software (National Institutes of Health, Baltimore, MD, USA).

### 2.8. Immunohistochemistry

The paraffin-based sections were dewaxed and rehydrated in xylene and ethanol. The paraffin sections were repaired under high pressure in EDTA buffer (pH 9.0) for 8 min to expose the antigens. At the condition of 37 °C, the sections were sequentially incubated with 3% hydrogen peroxide blocking agent for 15 min and goat serum for 40 min. Next, slices were incubated overnight at 4 °C with primary antibodies against RNF183 (1:1000 dilution), synaptopodin (1:1000 dilution), and PKM2 (1:2000 dilution). The sections were then sequentially incubated for 15 min with the corresponding secondary antibody and streptavidin-horseradish peroxidase (HRP) each at 37 °C. The sections were then incubated with biotinized secondary antibodies and streptavidin–horseradish peroxidase (HRP) complexes. For the in vitro experiments, HPC cells were cultured on coverslips and subsequently fixed with 4% paraformaldehyde for 15 min at room temperature. At 37 °C, the sections were sequentially incubated with 3% hydrogen peroxide blocking agent for 15 min and goat serum for 40 min. Next, slices were incubated overnight at 4 °C with primary antibodies against RNF183 (1:1000 dilution). The sections were then sequentially incubated for 15 min with the corresponding secondary antibody and streptavidin-horseradish peroxidase (HRP) each at 37 °C. The sections were then incubated with biotinized secondary antibodies and streptavidin–horseradish peroxidase (HRP) complexes. Finally, the positive results were observed with diaminobenzidine (DAB). Images were captured using a fluorescence microscope.

### 2.9. Immunofluorescence

The paraffin-based sections were dewaxed and rehydrated in xylene and ethanol. The paraffin sections were repaired under high pressure in EDTA buffer (pH 9.0) for 8 min to expose the antigens. Then, the sections were incubated with goat serum for 40 min at 37 °C. The sections were incubated overnight with specific antibodies (rabbit anti-RNF183 1:150 plus mouse anti-nestin 1:200, rabbit anti-PKM2 1:250 plus mouse anti-nestin 1:200) for 4 °C. Then, the sections were washed with PBS and incubated with fluorescent secondary antibodies and DAPI at 37 °C for 2 h. For the in vitro experiments, HPC cells were cultured on coverslips and subsequently fixed with 4% paraformaldehyde for 15 min at room temperature. Then, the sections were incubated with goat serum at 37 °C for 40 min and incubated overnight at 4 °C with specific antibodies (rabbit anti-calretin antibody, 1:2000). Then, the sections were washed with PBS and incubated with fluorescent secondary antibodies and DAPI at 37 °C for 2 h. Finally, images were captured using a fluorescence microscope.

### 2.10. RNA Immunoprecipitation-qPCR

The m6A RNA methylation levels were assessed following the protocol provided by the RNA Immunoprecipitation (RIP) Kit (Cat. No. P0102; Geneseed, Guangzhou, China). Briefly, the m6A antibody or Flag antibody was used to capture the target protein, to realize the enrichment of the target protein–RNA complex, and then to explore the intermolecular binding relationship through the detection and analysis of the protein and RNA in the complex. The RNA in the complex was isolated, reverse transcribed, and finally, m6A RNA levels were assessed using qPCR and quantified after standardized input. The m6A qPCR primers are primarily human RNF183 sense: 5′-CCCTTCAACAACACGTTCCAT-3′ and human RNF183 antisense: 5′-CGTGGGCAAGTCAGTGACAG-3′.

### 2.11. Co-Immunoprecipitation Assay

To clarify the relationship between RNF183 and PKM2, we collected the total protein of HPC cells and determined their concentration after up-regulating the expression of RNF183. Total protein was extracted from HPC cells, and its concentration was quantified using the BCA assay. Equal amounts of protein were then incubated with anti-Flag M2 affinity gel and rotated overnight at 4 °C. The samples were washed using centrifugation at 3000× *g* for 3 min, which was repeated five times. After collecting the sample precipitate, denaturation was performed.

### 2.12. LC–MS/MS Analysis

Following the co-immunoprecipitation (Co-IP) experiment, the samples to be tested were subjected to analysis using a Q Exactive mass spectrometer at Zhongke New Life Biotechnology Co., Ltd., Shanghai, China.

### 2.13. RNA Sequence

We isolated total RNA from mouse renal cortex tissues of *db/m* and *db/db* mice, respectively, and performed RNA sequencing at Meji Biomedical Technology Co., Ltd., Shanghai, China. Total RNA was purified, fragmented, and synthesized from the first chain cDNA. The second chain cDNA was then generated using PCR. After ligating the adapters to the ends of the cDNA fragments, they were sequenced on an IlluminaNovaSeq™ 6000 (Illumina, San Diego, CA, USA).

### 2.14. Statistical Analysis

All data were presented as means ± standard deviation (SD) and obtained from a minimum of three independent experiments. Statistical analysis was conducted using GraphPad Prism 8.0 software (GraphPad Inc., Boston, CA, USA). For comparisons between the two groups, Student’s *t*-test was applied. When more than two groups were involved, a one-way analysis of variance (ANOVA) was performed, followed by the Bonferroni post hoc test. The Kruskal–Wallis test was used for data that did not follow a normal distribution or exhibited variance heterogeneity. A *p*-value of less than 0.05 was considered statistically significant.

## 3. Results

### 3.1. Expression of RNF183 mRNA Levels Was Suppressed in the Kidney Cortex of db/db Mice

We detected differentially expressed genes in the kidney tissues of *db/db* and *db/m* mice using RNA sequencing, with green representing downregulated genes and red representing upregulated genes. As shown in Figure 1A and Appendix A, we found that the expression of RNF183 was significantly downregulated in the kidneys of *db/db* mice. Among the closely related genes with high homology to RNF183, RNA sequence results revealed significantly lower levels of RNF183 in *db/db* mice compared with *db/m* mice (Figure 1B). To confirm the RNA sequence results, quantitative real-time PCR was performed to quantify the levels of RNF183 mRNA in kidney tissues from *db/db* mice. The results showed that RNF183 mRNA levels in *db/db* mice were significantly lower than those in *db/m* mice (Figure 1C). The urine albumin-to-creatinine ratio (UACR) levels were significantly upregulated with *db/db* mice compared with *db/m* mice (Figure 1D).

### 3.2. In the Kidneys of Diabetic Mice, the Expression of RNF183 Was Found to Be Lower in the Podocytes

For the first time, we reported that the expression of RNF183 in glomerular podocytes from patients with DKD decreased with the course of the disease (Figure 2A). Next, we performed immunofluorescence double staining for RNF183 and the podocyte marker nestin. As shown in Figure 2B, RNF183 was co-expressed with nestin in kidney podocytes of DKD patients. The expression of RNF183 (red) and the podocyte marker nestin (green) was decreased in kidney podocytes of DKD patients compared with controls. It is noteworthy that among all subjects, the level of RNF183 in the glomerular was positively correlated with the estimated glomerular filtration rate (eGFR, Figure 2C) and negatively correlated with the urine protein-to-creatinine ratio (UPR, Figure 2D). As shown in Figure 2E, RNF183 was co-expressed with nestin in kidney podocytes of *db/db* mice. Compared with control mice, diabetic mice exhibited a reduced expression of RNF183 (red) and the podocyte marker nestin (green) in their kidney podocytes. Similarly, we also verified these findings with Western blot analysis, and the results agreed with the immunofluorescence staining results. Compared with control mice, the expression of RNF183 in the kidney tissue of diabetic mice was decreased (Figure 2F).

### 3.3. Reduced RNF183 Expression in High Glucose-Stimulated Podocytes

We investigated the alterations in both mRNA and protein expression of RNF183 in vitro in podocytes cultured with high glucose. As shown in Figure 3A, after 48 h of high glucose stimulation, the expression of RNF183 mRNA decreased. The results of the Western blot assay showed a time-dependent decrease in RNF183 protein expression with the prolonged duration of high glucose stimulation (Figure 3B). We transfected the overexpression of the RNF183 plasmid in podocytes cultured with normal glucose. Immunofluorescence double staining confirmed that RNF183 (green) colocalized with the ER marker calreticulin (red) (Figure 3C). Immunocytochemistry also confirmed that RNF183 expression was also decreased in podocytes after 48 h of high glucose treatment, expressed as brown granules (Figure 3D). As the duration of high glucose treatment increased, the protein levels of synaptopodin, BAX, BCL-2, LC3, and P62 were altered in a high glucose time-dependent manner in HPC cells. The Western blot assay results confirmed that the expression of the podocyte marker synaptopodin was reduced by 31% after 48 h of high glucose stimulation. Meanwhile, autophagy was inhibited (LC3-II/LC3-I ratio decreased by 42% and 1.74-fold increase in P62 expression increased by 1.74 times), and apoptosis was promoted (0.42-fold decrease in BCL-2/BAX ratio) (Figure 3E).

### 3.4. Downregulation of Rnf183 Was Involved in Autophagy and Apoptosis of HPC Cells Treated with High Glucose

To examine the role of RNF183 in the change in HPC cell function, we transfected plasmid PgenESIL-1-RNF183 or pGenesil-1 control plasmid into normal glucose-cultured HPC cells and observed the transfection efficiency. As shown in Figure 4A, the transfection efficiency was approximately 60%. After the transfection of the pGenesil-1-RNF183 plasmid in HPC cells, altered mRNA and protein expression of RNF183 were examined. The mRNA expression of RNF183 was significantly decreased by 0.14-fold (Figure 4B), and the protein expression was significantly decreased by 40.8% (Figure 4C). After autophagy inhibition in HPC cells, intracellular metabolic waste could not be properly excreted and continued to accumulate, leading to cell apoptosis. The autophagy flow experiment also confirmed that after transfection of the LC3-GFP-RFP plasmid, the number of autophagosomes (yellow fluorescence) increased significantly compared with the control group (Figure 4D), indicating that autophagy in HPC cells was inhibited after RNF183 knockdown. Next, Western blot assay results confirmed that podocyte marker (synaptopodin, Nephrin, and nestin) expression was decreased in HPC cells transfected with the plasmid pGenesil-1-RNF183 compared with control cells. Meanwhile, the suppression of autophagy in HPC cells was evident, as reflected by a 74% reduction in the LC3-II/LC3-I ratio, P62 expression increased to 1.68 times), and promoted apoptosis (BCL-2/BAX ratio decreased by 0.61 times) (Figure 4E). Subsequently, we examined whether the overexpression of RNF183 could mitigate the effects of high glucose on podocyte autophagy and apoptosis. HPC cells cultured in a high glucose medium were transfected with either the PCMV3-GFP-RNF183 overexpression plasmid or the PCMV3-GFP control vector. After transfection of the PCMV3-GFP-RNF183 overexpression plasmid in a high glucose medium, the transfection efficiency was approximately 90%, as assessed by the fluorescence images (Figure 4F). Real-time PCR assay results indicated that the cells upregulated by RNF183 were resistant to high glucose stimulation. The RNF183 mRNA levels were significantly upregulated, showing an increase by 67.4-fold compared with the cells transfected with PCMV3-GFP and treated with high glucose (Figure 4G). We transfected the LC3-GFP-RFP plasmid into high glucose HPC cells and found that the yellow fluorescence in HPC cells treated with PCMV3-GFP-RNF183 was significantly lower compared with the control group. This suggested that after the upregulation of RNF183 expression, the autophagic function of HPC cells was restored (Figure 4H). Furthermore, the expression of the podocyte marker Nephrin was increased in HPC cells of high glucose-treated PCMV3-GFP-RNF183 compared with high glucose-treated PCMV3-GFP transfected cells. Meanwhile, the LC3-II/LC3-I ratio increased, P62 was downregulated, and the BCL-2/BAX ratio also increased (Figure 4I).

### 3.5. m6A RNA Methylation May Play a Role in the Downregulation of RNF183 Induced by High Glucose in HPC Cells

To investigate the mechanism behind the decreased expression of RNF183 mRNA in podocytes under high glucose conditions, we treated the cells with three distinct drugs. To investigate the impact of various inhibitors on high glucose-induced changes in HPC cells, cells cultured in high glucose were randomly treated with DMSO, the histone deacetylase inhibitor trichostatin A (TSA), the RNA demethylase inhibitor meclofenamic acid 2 (MA2), or the DNA methyltransferase inhibitor 5-aza-2′-deoxycytidine (5-Aza) before being subjected to further high glucose stimulation. Western blot analysis demonstrated that treatment with MA2 led to a 3.05-fold increase in RNF183 protein levels compared with the DMSO-treated group (Figure 5A). RNA immunoprecipitation (RIP) -qPCR showed that the m6A antibody significantly enriched RNF183 mRNA compared with the immunoglobulin G (IgG) group. However, the m6A-specific antibodies significantly reduced RNF183 mRNA enrichment in high glucose-stimulated HPC cells (Figure 5B). To further determine the cause of the changes in m6A modification in HPC cells, we performed protein blotting on HPC cells incubated under high glucose conditions. As shown in Figure 5C, the expressions of METTL3, METTL14, and IGF2BP2 in HPC cells treated with high glucose decreased significantly at 48 h. Additionally, the expression of IGF2BP3 was downregulated in HPC cells cultured with high glucose for 36 h, with no significant changes observed in other enzymes. Among them, the changes in the m6A writer METTL3 and the reader IGF2BP2 were the most significant, suggesting that they may primarily mediate the downregulation of m6A modification levels in high glucose-induced HPC cells. The above results confirmed that RNF183 expression was regulated via m6A methylation modification, while METTL3 expression was decreased in response to high glucose. We speculated whether METTL3 would affect the expression of RNF183 mRNA, so real-time PCR experiments were performed. The results showed that the RNF183 transcript level was downregulated 0.58-fold after METTL3 knockdown, while the RNF183 protein level was also downregulated to some extent (Figure 5D,E). Meanwhile, our results also confirmed that the expression of IGF2BP2 was reduced upon high glucose stimulation. Considering that IGF2BP2 acts as a reader for m6A methylation and affects IGF2BP2 binding stability of downstream mRNA. We detected RNF183 alteration via IGF2BP2 knockdown, and the results showed that RNF183 was downregulated at both the transcription level and the protein level (Figure 5F,G). Furthermore, HPC cells cultured with normal glucose were transfected with PCMV3-Flag-IGF2BP2, and IGF2BP2 was detected using RIP–Western blot analysis. Meanwhile, RIP-qPCR showed a significant enrichment of RNF183 mRNA antibodies in the m6A group after IGF2BP2 upregulation compared with the IgG group (Figure 5H). We detected decreasing levels of m6A with disease progression in glomerular podocytes from patients with DKD (Figure 5I). We hypothesize that METTL3 and IGF2BP2 act synergistically to regulate RNF183, affecting the mRNA levels of RNF183, which subsequently leads to a decrease in RNF183 protein levels.

### 3.6. RNF183 Ubiquitinates PKM2 for Degradation

Given that RNF183 functions as an E3 ubiquitin ligase, we investigated whether it contributes to HPC cell damage by mediating the ubiquitination of substrate proteins. To identify the potential ubiquitination substrates of RNF183 in high glucose-induced HPC cell damage, we performed co-immunoprecipitation (Co-IP) combined with liquid chromatography–mass spectrometry (LC-MS) analysis. From these analyses, we first downregulated RNF183 under normal glucose conditions and performed LC-MS analysis on the whole-cell protein extracts. Among the differentially expressed proteins, we selected 343 proteins that were upregulated in the RNF183 knockdown group (Figure 6A and Appendix A). At the same time, after upregulating RNF183 under normal glucose conditions, we conducted Co-IP experiments combined with LC-MS analysis and selected 23 proteins that directly bound to RNF183 (Figure 6B and Appendix A). Based on the overlapping Venn plot of the two datasets, we screened two proteins, including pyruvate kinase M2 type pyruvate kinase (PKM2) and small ribosomal subunit protein (RACK1) (Figure 6C). Next, we validated the mass spectrometry results using Co-IP combined with Western blot assay experiments. The specific RNF183 vector PCMV3-Flag-RNF183 or the PCMV3-Flag control vector was transfected into HPC cells cultured with normal glucose in vitro. The results showed that RNF183 acts directly with PKM2 and RACK1 (Figure 6D). Using Western blot assay, we detected significant upregulation of PKM2 protein levels at 48 h, while RACK1 protein levels were not statistically significant (Figure 6E). As shown in Figure 6F, PKM2 was co-expressed with nestin in kidney podocytes of DKD patients. The expression of PKM2 (red) and the podocyte marker nestin (green) was increased in kidney podocytes of DKD patients compared with controls. To further confirm these findings, we investigated the expression levels of PKM2 and RACK1 proteins in the kidneys of *db/db* mice. The protein expression of PKM2 was significantly elevated in *db/db* mice, whereas no significant difference in RACK1 protein levels was observed when compared with control mice (Figure 6G). We also examined PKM2 expression in the kidneys of *db/db* mice using immunohistochemical staining. The levels of PKM2 expression were significantly higher in the glomerular podocytes of *db/db* mice compared with controls (Figure 6H). We performed immunofluorescence double staining for PKM2 and the podocyte marker nestin. As shown in Figure 6I, PKM2 and nestin were co-expressed in kidney podocytes of *db/db* mice. Compared with *db/m* mice, the expression of PKM2 (red) and the podocyte marker nestin (green) was increased in the podocytes of *db/db* mice.

### 3.7. RNF183-Mediated Downregulation of PKM2 Contributed to the Dysfunction of HPC Cells Treated with High Glucose

To investigate the underlying mechanism by which RNF183 facilitates the degradation of PKM2, HPC cells were treated with CHX, a protein synthesis inhibitor, and MG132, a proteasome inhibitor. Upon comparison with the MG132-treated group, the half-life of PKM2 was significantly shortened in CHX-treated HPC cells (Figure 7A). Furthermore, to assess the role of RNF183, we transfected HPC cells with either the pGenesil-1 control plasmid or the pGenesil-1-RNF183 plasmid for 48 h under normal glucose conditions. After transfection, CHX was added to HPC cells with RNF183 knockdown, and PKM2 levels were monitored over time. The data revealed that the half-life of PKM2 was notably extended in RNF183 knockdown HPC cells treated with CHX when compared with the control HPC cells (Figure 7B). Our mechanistic studies suggested that RNF183 induces the ubiquitination of PKM2, leading to a reduction in its protein levels. To verify the impact of RNF183’s targeting action on HPC cell function, we evaluated the substrate PKM2. We transfected HPC cells with the PCMV3-Flag-PKM2 plasmid and confirmed that overexpression of RNF183 in HPC cells altered PKM2 expression, thereby affecting HPC cell dysfunction. Successful upregulation of RNF183 expression in HPC cells cultured with high glucose was confirmed. PKM2 transcript levels were downregulated to 0.58-fold after RNF183 overexpression compared with the PCMV3-Flag group (Figure 7C). After transfecting high glucose-stimulated HPC cells with the PCMV3-Flag-RNF183 plasmid, we proceeded to introduce the PCMV3-Flag-PKM2 plasmid for further transfection. As expected, PKM2 protein levels were significantly upregulated. The PKM2 protein levels were downregulated to 0.17-fold after RNF183 overexpression when compared with the PCMV3-Flag group (Figure 7D). PKM2 protein levels were downregulated 0.11-fold compared with transcript levels, suggesting a post-transcriptional modification of the protein PKM2 by RNF183. In high glucose-treated HPC cells, the expression of the podocyte marker synaptopodin was significantly elevated in those co-transfected with PCMV3-Flag-RNF183 and PCMV3-Flag-PKM2 plasmids, compared with cells transfected with only PCMV3-Flag-RNF183. At the same time, a decrease in the LC3-II/LC3-I ratio was observed, accompanied by an increase in the expression of P62, and the BCL-2/BAX ratio increased (Figure 7E).

### 3.8. RNF183 Reduced PKM2 Expression Through the Ubiquitin–Proteasome Pathway

To further explore the site of PKM2 ubiquitination induced by RNF183, we made predictions through the protein online docking website (http://zdock.umassmed.edu/, Visit Date: 2 December 2024). According to the 3D structure map of the protein, the red-marked region is the site where RNF183 may play the role of E3 ubiquitin ligase and PKM2 (Figure 8A). A schematic representation of the structure of the pCI-neo-GFP-RNF183 plasmid mutant is shown in Figure 8B. To explore the type of PKM2 ubiquitination induced by RNF183, we co-transfected three mutated pCI-neo-GFP-RNF183 plasmids (WT, ΔTM, CS, and KR) with PCMV3-Flag-PKM2 into HPC cells treated with the proteasome inhibitor MG132. As shown via anti-Flag immunoblotting, the reduction in this modification was substantial when either the RING finger domain of RNF183 was mutated or the transmembrane domain was deleted (ΔTM) (Figure 8C). We then co-transfected the three mutant pCI-neo-GFP-RNF183 plasmids (WT, ΔTM, CS, and KR) with PCMV3-HA-Ub and PCMV3-Flag-PKM2 into HPC cells, followed by the addition of the proteasome inhibitor MG132. We determined the ubiquitination levels using anti-Flag immunoblotting. When the transmembrane domain of RNF183 was absent (Δ TM), this modification was greatly reduced (Figure 8D). In summary, these results confirmed that RNF183 mediated the ubiquitination and subsequent degradation of PKM2 in HPC cells.

## 4. Discussion

DKD is the leading cause of chronic kidney disease (CKD) and ESRD. One of the pathological features of DKD is the loss of glomerular podocytes and the crucial role that podocyte injury plays in the development of DKD. During DKD, podocytes become dysfunctional, primarily characterized by suppressed autophagy and accelerated apoptosis, which leads to elevated proteinuria and further kidney damage. In this study, we demonstrated, for the first time, that RNF183 expression in renal biopsy sections from DKD patients gradually decreased with the progression of diabetes. We also observed that RNF183 expression was significantly reduced in the kidneys of *db/db* mice and in HPC cells cultured with high glucose in vitro, which, in turn, led to cellular damage. However, the exact molecular mechanism underlying the downregulation of RNF183 expression in HPC cells induced via high glucose remains unclear and warrants further exploration. Moreover, the molecular mechanisms underlying reduced RNF183 expression also remain unclear and require further investigation. Therefore, we explored the relationship between RNF183 and podocyte dysfunction in the context of DKD. We used shRNA to knock down RNF183 in HPC cells in vitro, inhibiting autophagy and promoting apoptosis. Additionally, upregulating RNF183 expression in HPC cells cultured with high glucose successfully reversed the resulting cellular damage. This was evidenced by the increased expression of podocyte cytoskeletal proteins, enhanced autophagy, and reduced apoptosis. Therefore, targeting RNF183 for therapeutic intervention represents a promising approach to ameliorating DKD by restoring podocyte function in the glomerulus. Altered expression of RNF183 has been associated with a range of diseases, including CKD and inflammatory bowel disease, as well as various cellular processes, such as apoptosis and endoplasmic reticulum stress. In 2024, Ji et al. reported that knocking out RNF183 in FBXO5-deficient cells could reverse the apoptosis defect caused by FBXO5 deficiency in colorectal cancer cells [19]. In 2021, Li et al. demonstrated that upregulation of RNF183 expression induced the development of intestinal inflammation in a rat model of ulcerative colitis [20]. In 2022, Su et al. reported that in tumor cells, the inhibition of Cat D triggers NF-κB signaling activation through the degradation of autophagy-dependent IκB, which, in turn, increases RNF183 mRNA expression to induce apoptosis [21]. Therefore, we believe that the expression of RNF183 differed across tissues, and its role was different in various disease models.

Ubiquitination is a post-translational modification that is tightly regulated, involving the covalent bonding of ubiquitin molecules to the target proteins. The ubiquitination process is characterized by its intricate and dynamic nature, which governs a wide array of cellular functions. This includes the selective degradation of specific proteins, the repair of DNA damage, and the regulation of various cellular signaling pathways [22]. Ubiquitination is the cellular mechanism for protein degradation, which often occurs through the proteasome [23]. The ubiquitin–proteasome system relies on the coordinated action of three distinct classes of enzymes: E1, the ubiquitin-activating enzyme; E2, the ubiquitin-conjugating enzyme; and E3, the ubiquitin-ligating enzyme. Together, these enzymes orchestrate the attachment of ubiquitin molecules to target proteins, ultimately leading to their degradation [24,25]. Among these, E3 ligases play the most critical role in substrate recognition. There are three types of ubiquitin ligases, among which the ring finger (RNF) type is the most common [26]. The ring finger (RNF) protein family comprises a diverse group of proteins, many of which exhibit E3 ubiquitin ligase activity, often referred to as ring finger E3 ligases [27]. RNF183 is an E3 ubiquitin ligase that contains a RING finger and transmembrane domains, enabling it to specifically recognize substrates and carry out ubiquitination. In the process of podocyte injury, we investigated whether RNF183 could reduce protein levels by recognizing specific substrates, thereby protecting podocytes. In this study, we added the proteasome inhibitor cycloheximide (CHX) to inhibit protein synthesis and MG132 to block the proteasomal pathway in normal glucose-cultured cells. This resulted in an increase in PKM2 protein expression and a prolonged half-life. After knocking down RNF183 and treating the cells with CHX to inhibit protein synthesis, we observed that the half-life of PKM2 was extended. These findings suggested that PKM2 is a substrate of the E3 ubiquitin ligase RNF183 in podocytes. Rao et al. reported that in liver fibrosis, follistatin-like protein 1 (FSTL1) has been shown to interact directly with PKM2. Through its promotion of PKM2 phosphorylation and subsequent translocation to the nucleus, FSTL1 inhibits the ubiquitination of PKM2, which, in turn, helps to mitigate inflammation [28]. Wang et al. reported that the disruption of FBW7-mediated ubiquitin-dependent degradation of PKM2 in macrophages leads to significant alterations in redox balance, particularly exacerbating insulin resistance associated with obesity [29]. Subsequently, we examined whether RNF183 mediated the ubiquitination of PKM2, leading to its degradation. We also focused on identifying which domain of RNF183 mediated the ubiquitination. The results showed that this modification was significantly reduced when the transmembrane domain (ΔTM) of RNF183 was deleted. In podocytes, we examined whether RNF183 affected the expression of PKM2 protein, thereby influencing podocyte function. Additionally, we used functional rescue experiments to investigate whether RNF183 subsequently affects podocyte function by influencing the expression of PKM2 protein and whether PKM2 is involved in podocyte dysfunction. In high glucose-cultured HPC cells, the upregulation of PKM2 expression led to a significant exacerbation of cellular damage, manifested by reduced expression of podocyte cytoskeletal proteins, suppressed autophagy, as well as promoted apoptosis. Li et al.’s study showed that the increase in PKM2 triggered abnormal glycolysis in glomerular mesangial cells, leading to DKD fibrosis, which was reduced after recovery [30]. Previous studies have reported that ML-265-mediated PKM2 activation reduced entrance into the apoptosis cascade in in vitro and in vivo models of outer retinal stress [31]. Bailleul et al. found that in radiation-induced oxidative stress in glioblastoma (GBM), inhibiting the enzymatic activity of PKM2 increased glucose uptake, resulting in increased PPP flux [32]. These findings further confirmed that the accumulation of PKM2 causes exacerbated cell damage.

Transcription and protein levels of RNF183 were downregulated in HPC cells cultured with high glucose. We treated HPC cells with three epigenetic drugs and observed that treatment with the RNA demethylase inhibitor MA2 led to a significant increase in RNF183 expression under high glucose conditions. RIP experiments confirmed that RNF183 mRNA is significantly enriched by the m6A-specific antibody, indicating that RNF183 is regulated by m6A methylation. m6A RNA methylation modification is the most extensive RNA modification in eukaryotes; usually, the m6A modification occurs in the 3′-UTR region near the translation stop codon and is embedded in a conserved sequence 5′-RRACU-3 to achieve regulation of RNA by binding to a specific methylated reading protein [11]. The m6A methylation modification consists of three parts: the m6A writer (METTL3, METTL14, and WTAP), m6A eraser (ALKBH5 and FTO), and m6A reader (IGF2BP1/2/3) [13,14]. After high glucose stimulation of HPC cells, we examined the writers, erasers, and readers of m6A modifications and found significant changes in the expression of METTL3 and IGF2BP2. After knocking down METTL3 in HPC cells, both the mRNA and protein levels of RNF183 were significantly downregulated. Similarly, knocking down IGF2BP2 also led to a significant reduction in RNF183 expression. Considering that IGF2BP2, as an m6A reader, is an RNA-binding protein that regulates multiple biological processes, IGF2BP2 enhanced the stability and maintenance of its target mRNA in an m6A-dependent manner [33]. RIP experiments confirmed that IGF2BP2 maintains the stability of RNF183. We found that under high glucose conditions, the expression of METTL3 was downregulated, leading to a reduction in the m6A modification of RNF183. Additionally, the decreased expression of IGF2BP2 resulted in the reduced stability of RNF183 mRNA. METTL3 and IGF2BP2 worked synergistically to induce changes in the expression of RNF183. Previous studies reported that in podocytes of DKD, METTL3 regulated the Notch signaling pathway via m6A modifying TIMP2 in an IGF2BP2-dependent manner, which exerted pro-inflammatory and pro-apoptotic effects [34]. METTL3 and IGF2BP2, by influencing the expression of RNF183, subsequently caused changes in podocyte autophagy and apoptosis.

In HPC cells stimulated with high glucose, high glucose induced a decrease in METTL3 and IGF2BP2 levels, which mediated changes in the transcriptional levels of RNF183, leading to reduced RNF183 mRNA expression. Downregulated expression of RNF183 under high glucose conditions, leading to increased protein levels of PKM2, decreases autophagy and promotes apoptosis (Figure 9). The results of this study provide foundational insights for the clinical prevention and treatment of DKD and suggest that RNF183 holds promise as a novel therapeutic target and prognostic marker for DKD. The limitations of this study were as follows: First, although the mouse model provided valuable insights, extrapolation of the findings to human DKD patients may require further validation in clinical studies. Second, in conducting the investigation of molecular mechanisms, we performed and validated the mechanistic studies in vitro; however, we did not further validate them in gene-deficient mouse models, nor did we conduct any follow-up experiments in human cases of DKD.

## Figures and Tables

**Figure 1 cells-14-00365-f001:**
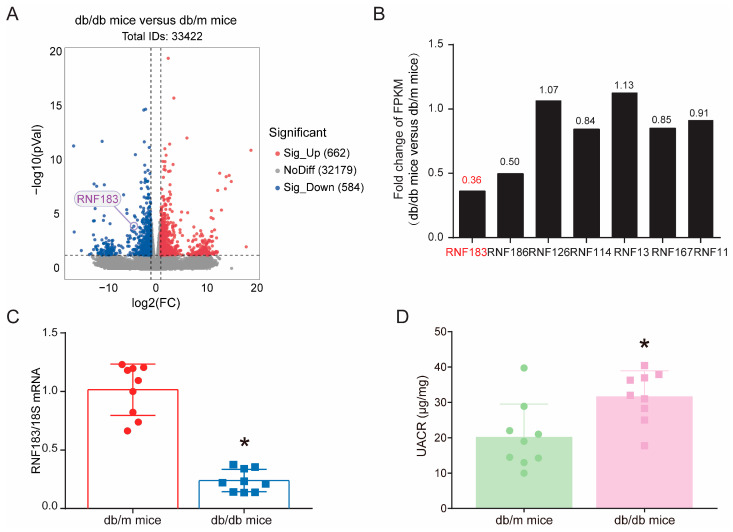
Expression of RNF183 mRNA levels was suppressed in the kidney cortex of *db/db* mice. (**A**) Differential gene volcano map of the RNA-seq results, |log2 (fold change)| ≥ 1, and *p* < 0.05. (**B**) Genes that are closely related with high homology within the RNF family. (**C**) Changes in RNF183 mRNA expression in mouse kidney tissues were determined using real-time PCR. * *p* < 0.05 versus *db/m* mice. (**D**) UACR levels in *db/m* and *db/db* mice. * *p* < 0.05 versus *db/m* mice.

**Figure 2 cells-14-00365-f002:**
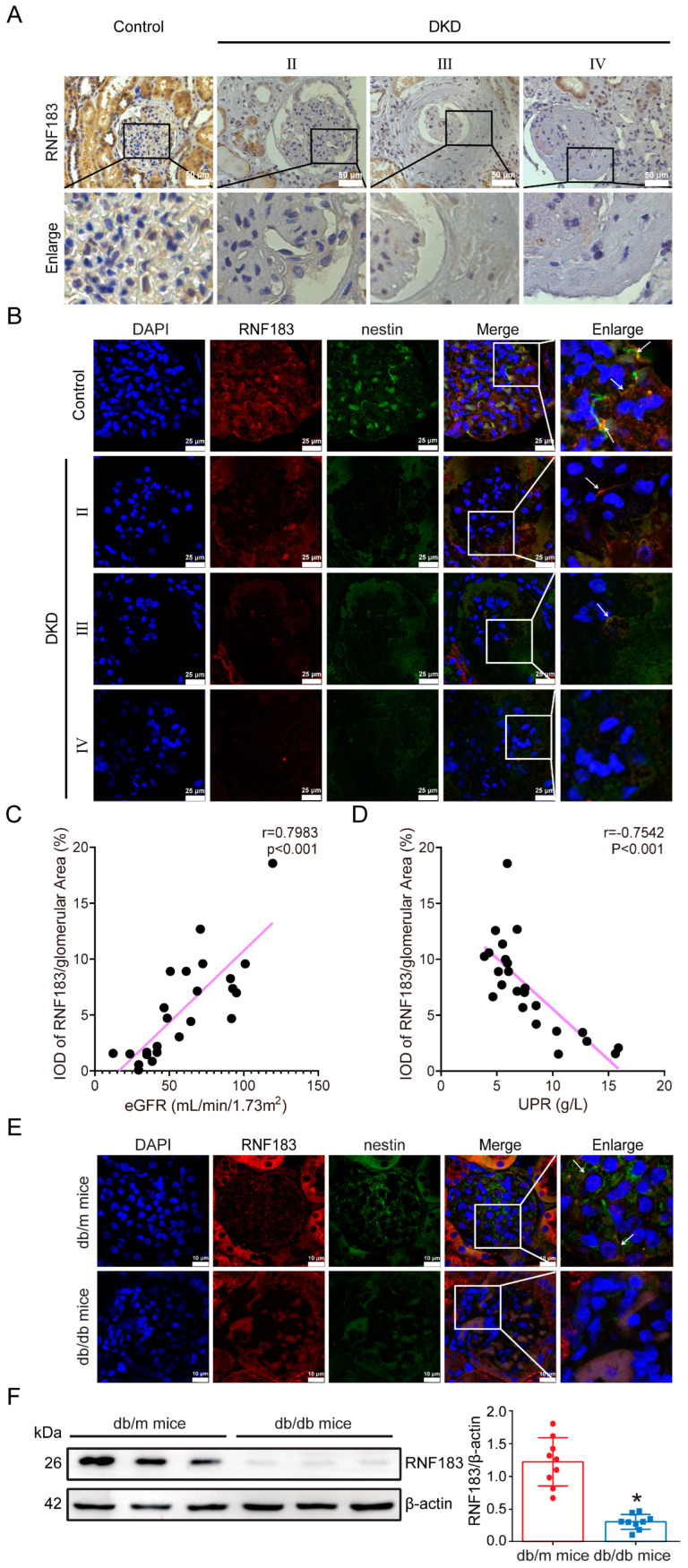
In the kidneys of diabetic mice, the expression of RNF183 was found to be lower in the podocytes. (**A**) Immunohistochemical results of RNF183 in renal tissue of DKD II stage, III stage, and IV stage patients. (**B**) The colocalization of RNF183 with the podocyte marker nestin in the human kidney tissues from each group was detected using immunofluorescence staining (White arrow indicated the positive area). (**C**) Correlation between renal RNF183 and eGFR in patients with DKD. (**D**) Correlation between renal RNF183 and UPR in patients with DKD. (**E**) The colocalization of RNF183 with the podocyte marker nestin in the kidney tissues of mice from each group was detected using immunofluorescence staining (White arrow indicated the positive area). (**F**) The expression of RNF183 protein in renal tissues of each group was detected using Western blot analysis. * *p* < 0.05 versus *db/m* mice.

**Figure 3 cells-14-00365-f003:**
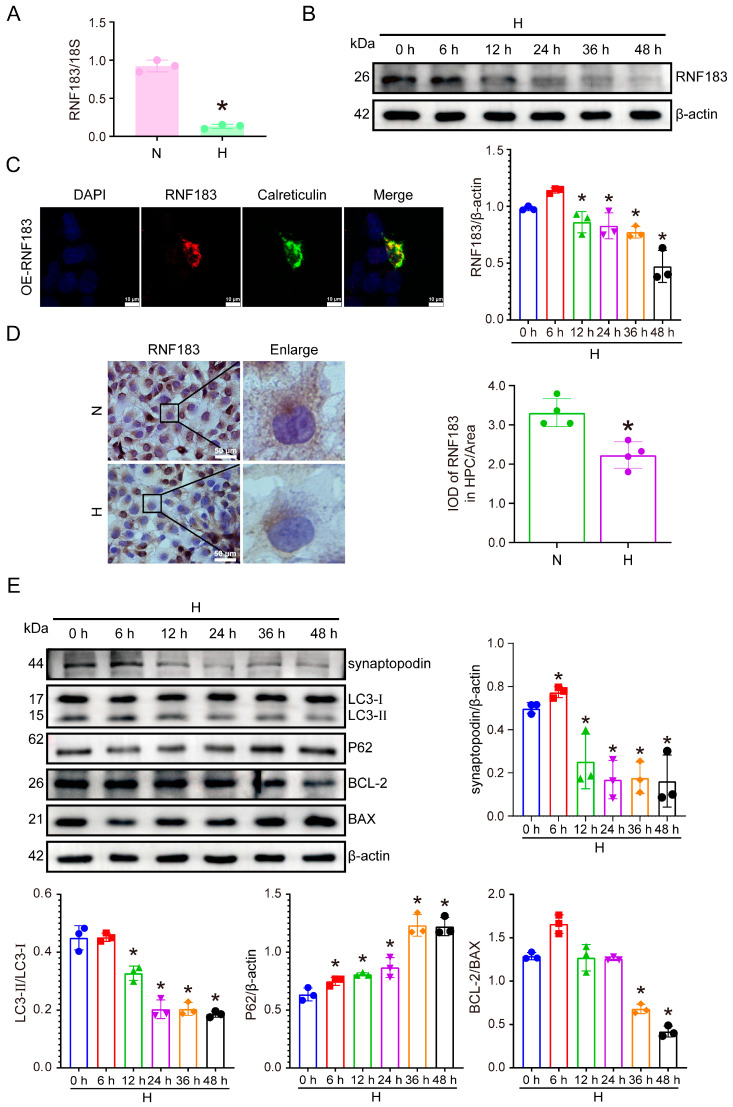
Decreased RNF183 expression in high glucose-stimulated podocytes is accompanied by functional alterations. (**A**) Real-time PCR was employed to measure the mRNA expression of RNF183 in HPC cells subjected to high glucose treatment. * *p* < 0.05 versus the N group. (**B**) Western blot analysis was performed to assess RNF183 protein expression in high glucose-treated HPC cells at different time points. * *p* < 0.05 versus the N group. (**C**) Colocalization of RNF183 and calreticulin was detected using immunofluorescence double staining in HPC cells. (**D**) Immunocytochemical analysis of RNF183 expression in high glucose-treated HPC cells. (**E**) Western blot analysis was performed to assess the protein expression of synaptopodin, BAX, BCL-2, LC3, and P62 in HPC cells treated with high glucose time points. * *p* < 0.05 versus the N group.

**Figure 4 cells-14-00365-f004:**
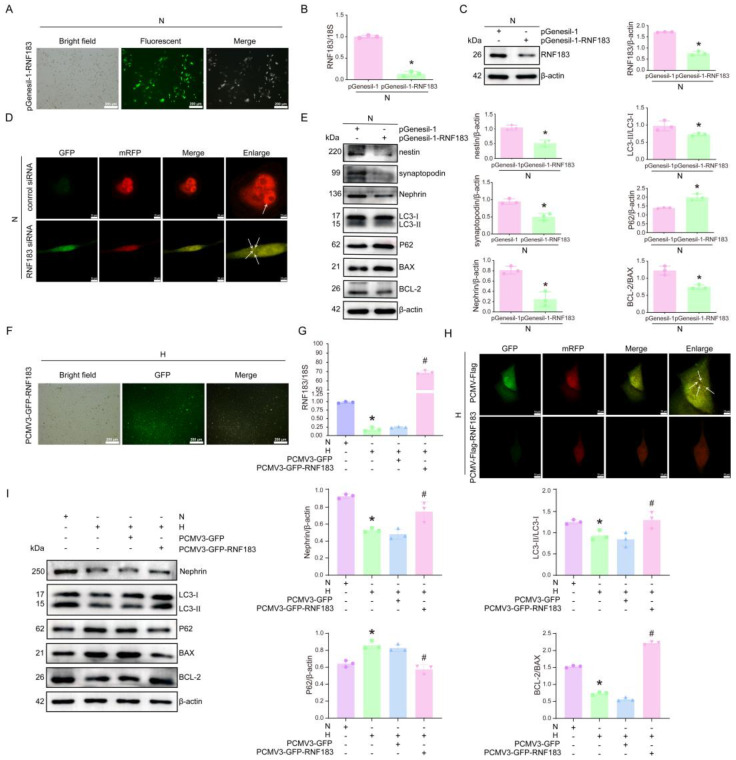
Downregulation of Rnf183 was involved in autophagy and apoptosis of HPC cells treated with high glucose. (**A**) Transfection efficiency was evaluated by comparing bright field, fluorescence field, and merged images of HPC cells transfected with pGenesil-1-RNF183. (**B**) Changes in RNF183 mRNA expression in subsequent HPC cells, as measured using real-time PCR. * *p* < 0.05 versus the pGenesil-1 group. (**C**) Western blot assay was performed to evaluate the expression of RNF183 in HPC cells following transfection with pGenesil-1-RNF183. * *p* < 0.05 versus the pGenesil-1 group. (**D**) Fluorescence images of the LC3-GFP-RFP tandem probe were obtained by scanning different channels using a confocal microscope (White arrow indicated the positive area). (**E**) Western blot analysis was performed to assess the protein expression levels of nestin, synaptopodin, Nephrin, LC3, P62, BAX, and BCL-2 in HPC cells following RNF183 knockdown. A statistically significant difference was observed compared with the pGenesil-1 control group. * *p* < 0.05 versus the pGenesil-1 group. (**F**) Transfection efficiency was assessed by comparing bright field, fluorescence, and merged images of HPC cells transfected with PCMV3-GFP-RNF183. (**G**) The expression of RNF183 was detected using real-time PCR after transfection of the PCMT3-GFP-RNF183 plasmid into HPC cells treated with high glucose. * *p* < 0.05 versus the N group; # *p* < 0.05 versus the H plus PCMV3-GFP group. (**H**) Fluorescence images of the LC3-GFP-RFP tandem probe were obtained by scanning with different channels using confocal microscopy (White arrow indicated the positive area). (**I**) Changes in the protein levels of Nephrin, LC3, P62, BAX, and BCL-2 were examined using Western blot analysis, following the overexpression of RNF183 in HPC cells cultured under high glucose conditions. * *p* < 0.05 versus the N group; # *p* < 0.05 versus the H plus PCMV3-GFP group.

**Figure 5 cells-14-00365-f005:**
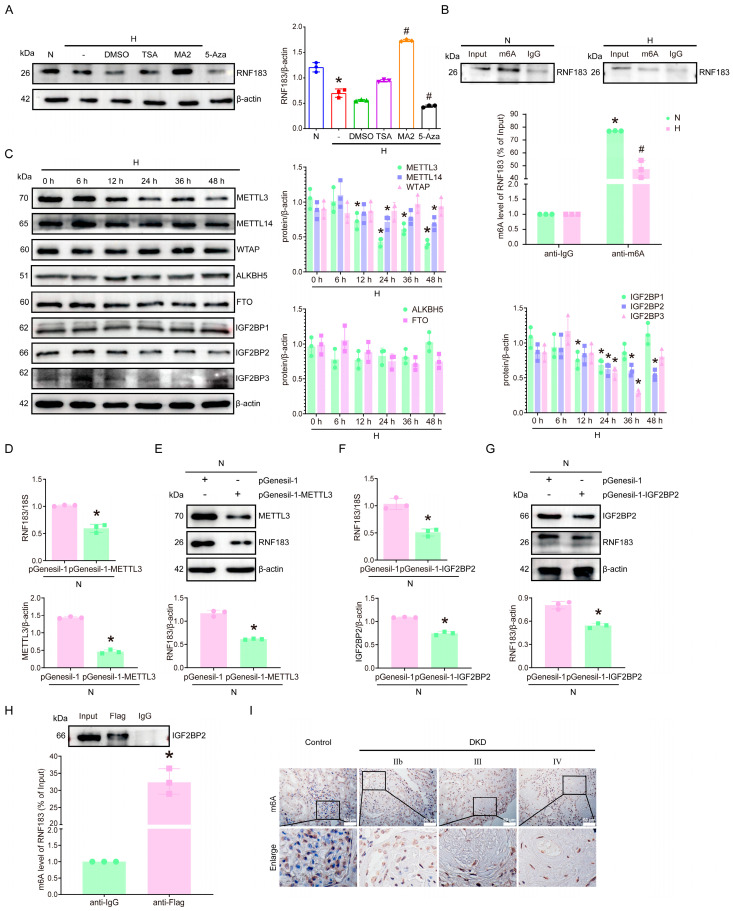
m6A RNA methylation may play a role in the downregulation of RNF183 induced via high glucose in HPC cells. (**A**) Changes in RNF183 protein expression in HPC cells induced by high glucose were measured using Western blot analysis. * *p* < 0.05 versus the N group; # *p* < 0.05 versus the H plus DMSO group. (**B**) RIP-qPCR analysis indicated m6A modification of RNF183 in HPC cells treated with high glucose. The level of m6A modification of RNF183 was reduced in HPC cells cultured with high glucose. * *p* < 0.05 versus the anti-IgG group; # *p* < 0.05 versus the N group. (**C**) Expression of genes involved in RNA m6A methylation regulation in HPC cells cultured with high glucose was determined using Western blot analysis. * *p* < 0.05 versus the 0 h group. (**D**) The mRNA expression of METTL3 and RNF183 was detected using real-time PCR after transfection of pGenesil-1-METTL3 in HPC cells. * *p* < 0.05 versus the pGenesil-1 group. (**E**) Changes in METTL3 and RNF183 protein expression after METTL3 knockdown in HPC cells were determined using Western blot analysis. * *p* < 0.05 versus the pGenesil-1 group. (**F**) Changes in IGF2BP2 and RNF183 mRNA expression after IGF2BP2 knockdown in HPC cells were determined using real-time PCR. * *p* < 0.05 versus the pGenesil-1 group. (**G**) Western blot analysis of changes in IGF2BP2 HPC cells and RNF183 protein expression following IGF2BP2 knockdown. * *p* < 0.05 versus the pGenesil-1 group. (**H**) The enrichment of RNF183 m6A modification using IGF2BP2 binding was determined using RIP-qPCR with Flag-specific antibodies and IgG control antibodies. * *p* < 0.05 versus the anti-IgG group. (**I**) Immunohistochemical results of m6A in the renal tissue of DKD patients.

**Figure 6 cells-14-00365-f006:**
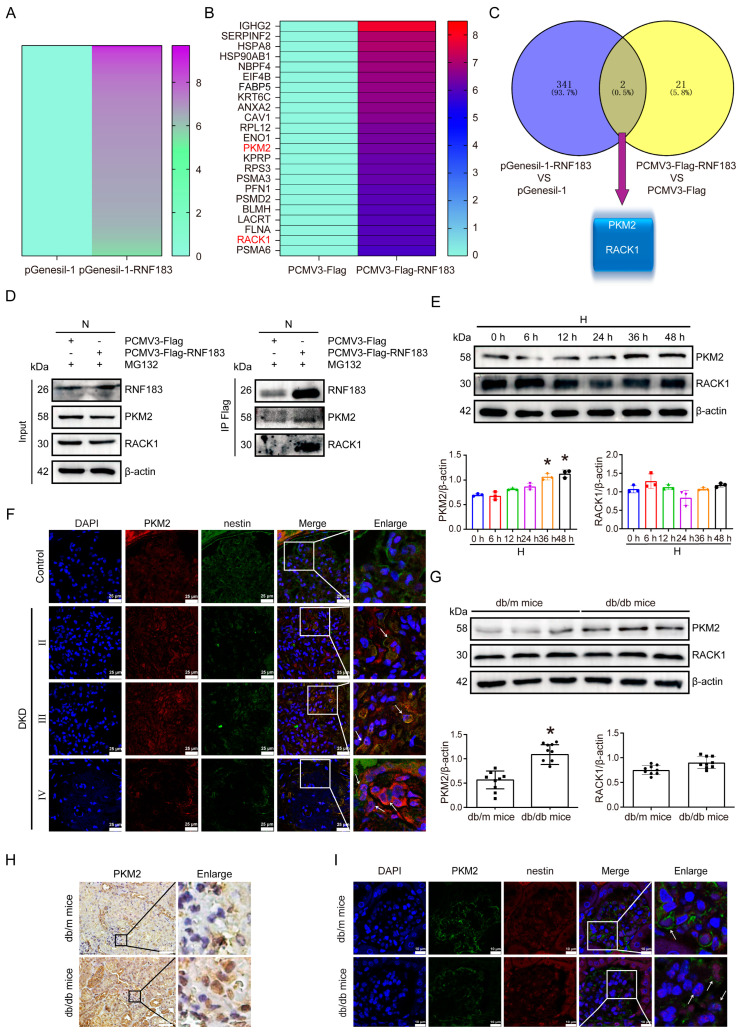
RNF183 ubiquitinates PKM2 for degradation. (**A**) HPC cells were transfected with either the pGenesil-1-RNF183 plasmid or the pGenesil-1 plasmid as a control group, and the whole-cell protein extracts of HPC cells were detected by LC/MS omics. (**B**) Co-IP experiments were performed after transfecting of HPC cells with plasmids, PCMV3-Flag-RNF183 and PCMV3-Flag, and the proteins in HPC cells were identified using LC/MS omics. (**C**) A Venn plot was obtained based on the overlap of the two datasets, results A and B https://bioinfogp.cnb.csic.es/tools/venny/index.html “URL (Visit Date: 18 November 2022)”. (**D**) Co-IP experiments examined the direct interaction of RNF183 with PKM2 or RACK1. Cell lysates were precipitated with anti-Flag M2 affinity gel and subjected to IB with the indicated antibodies. (**E**) Western blot analysis of changes in PKM2 or RACK1 protein expression in HPC cells cultured with high glucose at different time points. * *p* < 0.05 versus the N group. (**F**) The colocalization of PKM2 with the podocyte marker nestin in the human kidney tissues from each group was detected using immunofluorescence staining (White arrow indicated the positive area). (**G**) Changes in PKM2 or RACK1 protein expression in kidney tissues from each group, as determined using Western blot analysis. * *p* < 0.05 versus the N group. (**H**) Immunohistochemical analysis of PKM2 expression in kidney tissues from each group of mice. (**I**) The colocalization of PKM2 with the podocyte marker nestin in the kidney tissues of mice from each group was detected using immunofluorescence staining (White arrow indicated the positive area).

**Figure 7 cells-14-00365-f007:**
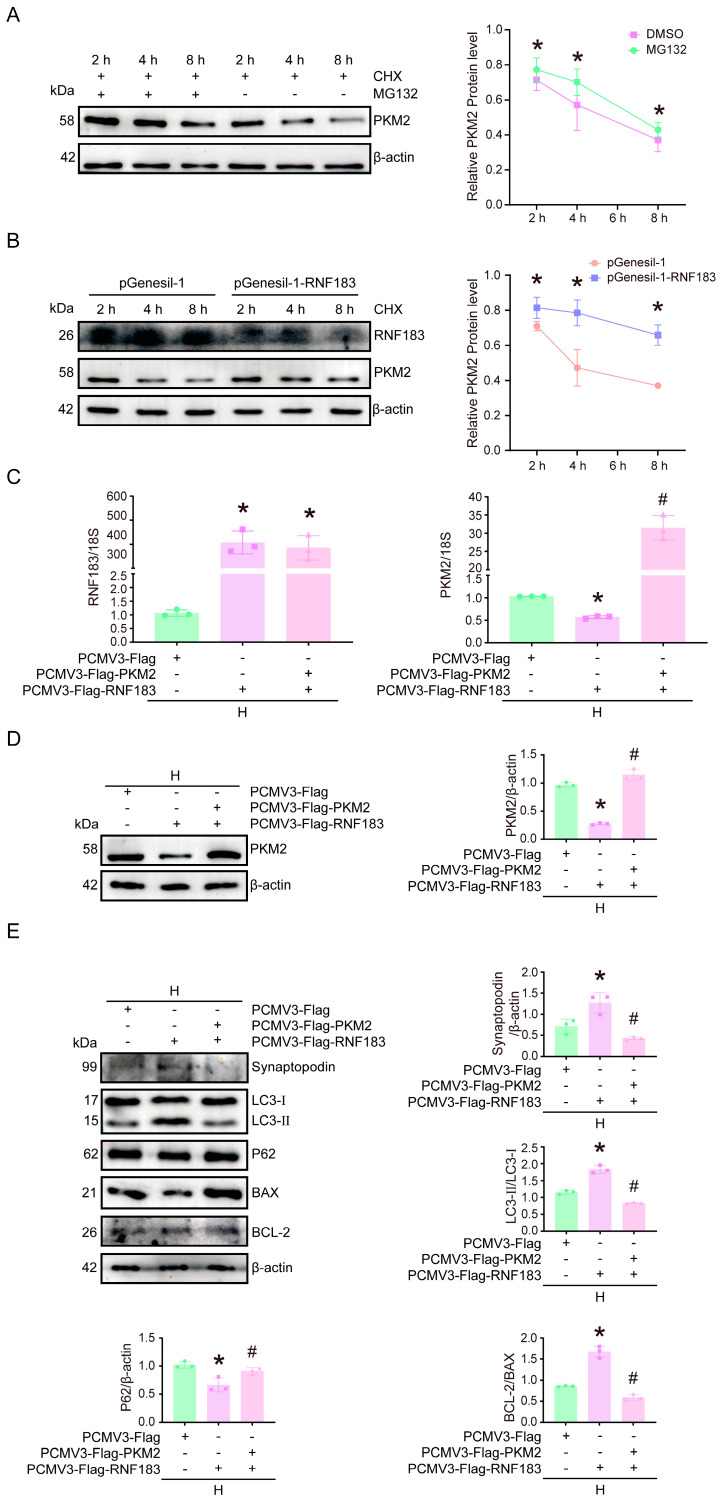
RNF183-mediated downregulation of PKM2 contributed to the dysfunction of HPC cells treated with high glucose. (**A**) Western blot analysis was conducted to assess PKM2 protein expression following MG132 treatment in CHX-treated HPC cells. * *p* < 0.05 versus the MG132 group. (**B**) CHX was added after transfection with plasmid pGenesil-1 or pGenesil-1-RNF183 in HPC cells, and Western blot analysis was conducted to evaluate the expression levels of RNF183 and PKM2 proteins. * *p* < 0.05 versus the pGenesil-1 group. (**C**) HPC cells were cultured under high glucose conditions and then transfected with PCMV3-Flag, PCMV3-Flag-RNF183, PCMV3-Flag-PKM2, or PKM2 overexpression plasmids, and changes in RNF183 or PKM2 mRNA expression were detected using quantitative real-time PCR. * *p* < 0.05; # *p* < 0.05 versus the high glucose + PCMV3-Flag group. * *p* < 0.05; # *p* < 0.05 compared with the high glucose plus PCMV3-Flag group. (**D**) HPC cells cultured in high glucose were transfected with PCMV3-Flag, PCMV3-Flag-RNF183, PCMV3-Flag-PKM2, or PKM2 overexpression plasmids, and changes in RNF183 protein expression were measured using Western blot assay. * *p* < 0.05; # *p* < 0.05 compared with the high glucose + PCMV3-Flag group. * *p* < 0.05; # *p* <0.05 versus the high glucose plus PCMV3-Flag group. (**E**) HPC cells cultured under high glucose conditions were transfected with PCMV3-Flag, PCMV3-Flag-RNF183, or PCMV3-Flag-PKM2, and the protein expression levels of synaptopodin, LC3, P62, BAX, and BCL-2 were determined using Western blot assay. * *p* < 0.05 versus the high glucose plus PCMV3-Flag group; # *p* < 0.05 versus the high glucose plus PCMV3-Flag-RNF183 group.

**Figure 8 cells-14-00365-f008:**
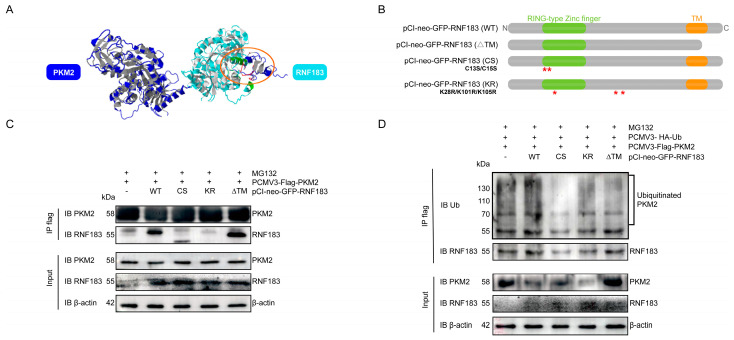
RNF183 reduced PKM2 expression through the ubiquitin–proteasome pathway. (**A**) Three-dimensional structure diagram predicting the docking of RNF183 and PKM2 proteins (http://zdock.umassmed.edu/, Visit Date: 2 December 2024). (**B**) The picture shows a schematic representation of the RNF183 structure. The red asterisk (*) indicates the amino acids mutated in the RNF183 mutant. The red asterisk (*) indicates the 13th and 15th amino acid from cysteine (Cysteine, C) to serine (Serine, S) and 28th, 101th and 105th amino acid from lysine (Lysine, K) to arginine (Arginine, R) in RNF183. (**C**) HPC cells were treated with MG132 and co-transfected with PCMV3-Flag-PKM2 and pCI-neo-GFP-RNF183, as detected using Co-IP experiments. Cell lysates were precipitated with anti-Flag M2 affinity gel and analyzed using immunoblotting (IB) with the indicated antibodies. (**D**) HPC cells were treated with MG132 and co-transfected with PCMV3-HA-Ub, PCMV3-Flag-PKM2, and pCI-neo-GFP-RNF183, and Co-IP experiments were performed for PKM2 and RNF183. The cell lysates were precipitated with anti-Flag M2 affinity gel and treated with specific antibody IB.

**Figure 9 cells-14-00365-f009:**
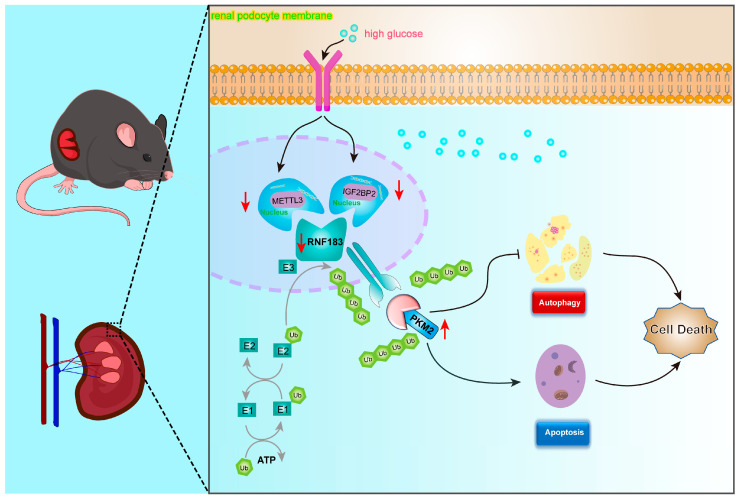
A model of the high glucose regulation of m6A, RNF183, PKM2, autophagy, and apoptosis in glomerular podocytes.

## Data Availability

All data generated in this study are available upon reasonable request from the corresponding author.

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
