# Peer review of "Modification of RNF183 via m6A Methylation Mediates Podocyte Dysfunction in Diabetic Nephropathy by Regulating PKM2 Ubiquitination and Degradation"

_cells, 2025, doi:10.3390/cells14050365_

Round 1
Reviewer 1 Report
Comments and Suggestions for Authors
Here the authors show reduced RNF183 expression in diabetic kidney biopsies as well as reduced expression in podocytes under hyperglycemia in vitro, and in diabetic mouse kidneys. In addition, the authors show that RNF183 is regulated by m6A methylation.
The study is overall interesting, and many methodologies were used that is impressive.
However, several concerns arise while reading the draft.
Major points:
1) The description of methodologies is somewhat superficial, contains many errors or lacks basic information that allows transparency and reproducibility.
2) In line 119, "RNF183 was purchased" presumably was not the recombinant protein. Why is its catalogue number same as for Nestin?
3) Line 173: how were mice divided randomly into 2 groups by genotype if there is already 2 different genotype groups? You can not mix db/db and db/m groups for randomization...
4) Line 175: None of the plasmid constructs are described. What was in the construct? What was the vector? What is the sequence?
5) Line 178: Describe transfection procedure in detail (transfection reagent dilution, manufacturer).
6) Line 183: Number of biological and technical replicates should be indicated.
7) Line 184: HPC cell source (original) should be clearly described. Also, podocyte cell culture needs special environment, this should be clearly described (eg thermosensitive cells?).
8) Line 185: 1640 medium is correctly RPMI-1640.
9) Line 224: How was protein extracted from kidneys and cells? Describe protein extraction reagents! Dilution of primary antibodies is missing.
10) Line 237 and line 248: For how long was HIER performed?
11) Line 268: What primers were used for the m6A qPCR?
12) Line 297: The used statistical analysis is incorrect, as ANOVA should not be used for non-parametric results. Student-Newman test is also inappropriate, with vary low power, Mann-Whitney shoud be used instead. Please re-calculate ANOVA statistics with appropriate non-parametric tests (Kruskal-Wallis). Post-hoc test should be described as well.
13) For all figures: bar charts should always include individual data points as scatter plot. Some of the charts inlcudes them, but some do not. Please correct.
14) Figure 2A: The RNF183 stain is not evidently localized to podocytes. A double immunofluorescence should be performed with a podocyte marker (eg synaptopodin).
15) Figure 2B: Why nestin was used to label podocytes? Synaptopodin is a much better marker, besides the green nestin labeling shows lot of extraglomerular false positive stain. I would repeat the stain with more convincing podocyte marker.
16) Figure 2C: This data does not add anything to Figure 2B, it is just a repetition. Besides, Synaptopodin staining separately from RNF183 is meaningless, only shows podocyte loss in diabetic mice.
17) Figure 4 is so small in the maunscript that is is impossible to evaluate properly.
18) Figure 5C: The western blot picture shows lots of bands with similar or very close molecular weight. How were all these proteins detected, if only one beta-actin band is shown? Were the blots re-probed 6-7 times??? Same question for blots of BCL2/BAX/LC3.
19) Figure 6G: To show convincingly that PKM2 plays a role in the diabetic podocytes, double immunofluorescent staining should be performed for exact protein localization. The photomicrograph shows massive tubular PKM2 expression in db/db kidneys.
20) PKM2 and m6A expression should be presented in the human kidney biopsies as well.
21) Demographic and basic clinical data of renal biopsy patietns should be added, including histology diagnosis, CKD staging , eGFR, proteinuria, medications.
22) Study limitations need to be added before the conclusions.
23) A 38% match with other publications, accorring to plagiarism report (iThenticate) should be corrected, extensive revision of the draft is needed (please re-phrase those "copy/paste" sentences).
Minor point: The many typos need to be corrected.
Comments on the Quality of English LanguageEnglish grammar is very poor in some subchapters and in the abstract, an extensive language editing is needed.
Author Response
Reviewer #1:
We would like to express our sincere gratitude to the reviewers for their meticulous review and constructive feedback, which has given us the opportunity to improve our manuscript in various aspects. We have carefully addressed each comment and have made modifications accordingly. And here we have marked in red in the revised paper.
1) The description of methodologies is somewhat superficial, contains many errors or lacks basic information that allows transparency and reproducibility.
Response: We sincerely thank you for your careful reading and for pointing out our mistakes. Following your suggestion, we have added a detailed description in manuscript place 2. Materials and Methods and corrected the errors.
2) In line 119, "RNF183 was purchased" presumably was not the recombinant protein. Why is its catalogue number same as for Nestin?
Response: Firstly I am sorry for the negligence and mistake. I have corrected the catalog number for the RNF183 (ARP43404) and nestin (sc-23927) antibodies. You can check it out on page 4 of the manuscript, lines 182-186.
- Line 173: how were mice divided randomly into 2 groups by genotype if there is already 2 different genotype groups? You can not mix db/db and db/m groups for randomization...
Response: Thanks so much for your friendly correction. In our revised manuscript 2.3. Animals, the errors have been revised as well as a detailed description of rat-related information. You can view it at lines 255-274 on page 7 of the manuscript.
4) Line 175: None of the plasmid constructs are described. What was in the construct? What was the vector? What is the sequence?
Response: Sorry to make the inconvenience. We made revisions in lines 277-293 on page 7 of the manuscript. Detailed experimental steps are as follows: The target sequence of the RNF183 gene (5′-CCACTCTTTGAGGGAGTGTTT-3′) was designed from the website https://www.sigmaaldrich.cn/CN/zh, and the corresponding single-stranded DNA oligonucleotides for the RNF183 shRNA were chemically synthesized. This oligonucleotide was annealed to form a double-stranded DNA fragment. The pGenesil-1 plasmid was digested with BamHI and HindIII restriction enzymes at 37°C for 3 hours. After gel extraction of the products, T4 ligase was used to ligate the products with the double-stranded DNA fragment at 16°C overnight. The recombinant plasmid was then transformed into Escherichia coli DH5α strain for amplification, and plasmid DNA was extracted using the TIANGEN kit. Due to the presence of a SalI recognition site in the single-stranded DNA oligonucleotide, a 400 bp fragment was obtained after SalI digestion. The successfully constructed plasmid was named pGenesil-1-RNF183.
5) Line 178: Describe transfection procedure in detail (transfection reagent dilution, manufacturer).
Response: Thank you for your suggestion. We have supplemented the transfection procedure in detail in lines 293 – 305 on page 7 of the manuscript. Detailed experimental steps are as follows: HighGene transfection reagents were purchased from ABclonal Biotechnology Co., Wuhan, Hubei, China. Cell transfection was carried out following the guidelines provided by the manufacturer for the HighGene transfection reagent. The detailed operation steps are as follows: We vortexed the plasmid (4 μg), serum-free RPMI-1640 (200 μl), and transfection reagent (8 μl) and incubated the mixture at room temperature for 15 minutes. Finally, the compound was added evenly dropwise into the wells of 6-well cell culture plates. After 6 h of transfection, half of the transfection medium was replaced with the specified medium, and the corresponding detection was performed after continued culture for 48 h.
6) Line 183: Number of biological and technical replicates should be indicated.
Response: Thank you for your advice. We have supplemented this in lines 366-367 on page 9 of the manuscript. All our cell experiments were greater than or equal to 3 independent replicate experiments.
- Line 184: HPC cell source (original) should be clearly described. Also, podocyte cell culture needs special environment, this should be clearly described (eg thermosensitive cells?).
Response: Thank you for suggesting that we add more detailed explanations in the 2.5. Cell culture and groups. We have supplemented therelevant sections based on your feedback, aiming to provide readers with amore comprehensive understanding of our methods. Details can be viewed in lines 307-327 on pages 7-8 of the manuscript.
8)Line 185: 1640 medium is correctly RPMI-1640.
Response: We were really sorry for our careless mistakes. Thank you for your reminder. We have changed the 1640 medium to the RPMI-1640. Details can be viewed in line 322 on page 8 of the manuscript.
9) Line 224: How was protein extracted from kidneys and cells? Describe protein extraction reagents! Dilution of primary antibodies is missing.
Response: Sorry to make the inconvenience. Thanks so much for your friendly correction. We have provided the company and product numbers of the RIPA lysis buffer in lines 209-210 on page 5 of the manuscript. We also supplemented details of the protein extraction step and primary antibody dilution in lines 386 – 389 and 396 – 405 on page 9 of the manuscript.
10)Line 237 and line 248: For how long was HIER performed?
Response: Thanks so much for your friendly suggestions. We have added the details on page 10, section 2.8, lines 415-416 and section 2.9, lines 429-430.
11)Line 268: What primers were used for the m6A qPCR?
Response: Thank you for your suggestions. Our m6A qPCR primers are primarily human RNF183 sense: 5′ - CCCTTCAACAACACGTTCCAT-3′ and human RNF183 antisense: 5′ - CGTGGGCAAGTCAGTGACAG-3′. You can view the details on page eleven, Section 2.10, lines 455-458.
12) Line 297: The used statistical analysis is incorrect, as ANOVA should not be used for non-parametric results. Student-Newman test is also inappropriate, with vary low power, Mann-Whitney shoud be used instead. Please re-calculate ANOVA statistics with appropriate non-parametric tests (Kruskal-Wallis). Post-hoc test should be described as well.
Response: We are very sorry for making the wrong description. Thank you for your reminder. We have revised the manuscript for error on page 11, lines 486-495.
13) For all figures: bar charts should always include individual data points as scatter plot. Some of the charts inlcudes them, but some do not. Please correct.
Response: Thank you for your friendly suggestion. Bar graphs are shown in our results, where each mouse from the in vivo experiment was used as an independent data point. However, for the cell experiments, we only performed three independent repeats, and the results were quantified, statistically analyzed, and presented without treating each data as an independent data point.
14) Figure 2A: The RNF183 stain is not evidently localized to podocytes. A double immunofluorescence should be performed with a podocyte marker (eg synaptopodin).
Response: Thanks so much for your critical advice. We are very sorry for the high false positive level of our results. We repeated the experiment by replacing nestin (sc-23927) with a better specific primary antibody nestin (ab6142). We have replaced the new results on page 14 of Fig. 2A of the manuscript. We replaced nestin (sc-23927) with the better specific primary antibody nestin (ab6142), with the new results available in Fig. 2A on page 14 of the manuscript. We have complemented the double immunofluorescence experiments of RNF183 with the podocyte marker nestin. You can view the results in Fig. 2B on page 14 of the manuscript.
15) Figure 2B: Why nestin was used to label podocytes? Synaptopodin is a much better marker, besides the green nestin labeling shows lot of extraglomerular false positive stain. I would repeat the stain with more convincing podocyte marker.
Response: Thank you for your constructive suggestion. Podocytes rely on their finger-like foot processes to attach to the surface of the capillary basement membrane, where the foot processes form convoluted filtration slits. These slits represent the final barrier of the glomerular filtration membrane, and the completion of this filtration function depends on the podocyte's robust cytoskeletal system. Nestin, a cytoskeletal protein belonging to the intermediate filament family, is expressed in highly differentiated podocytes and plays a crucial role in maintaining the normal morphology and function of podocytes. Nestin is abundantly expressed in the mature podocytes of both human and mouse kidneys. This problem was greatly reduced by replacing nestin (sc-23927) with a better specific primary antibody nestin (ab6142), which is corrected in Fig. 2E. You can view the results in Fig. 2E on page 14 of the manuscript.
16) Figure 2C: This data does not add anything to Figure 2B, it is just a repetition. Besides, Synaptopodin staining separately from RNF183 is meaningless, only shows podocyte loss in diabetic mice.
Response: Thanks for your suggestions in the results section. We have carefully adopted your advice and have made modifications accordingly. We have removed the results of Fig. 2C in the manuscript.
- Figure 4 is so small in the maunscript that is is impossible to evaluate properly.
Response: Thank you for pointing out our mistakes. For the experimental results of Figure 4, we really cannot intuitively calculate the accurate transfection efficiency. We revised this in lines 613 and 641 on page 17 of the manuscript.
18) Figure 5C: The western blot picture shows lots of bands with similar or very close molecular weight. How were all these proteins detected, if only one beta-actin band is shown? Were the blots re-probed 6-7 times??? Same question for blots of BCL2/BAX/LC3.
Response: Thank you very much for your kind guidance. For bands with similar or very close molecular weight, although we did not show the corresponding beta-actin for each band in the figures, we used their respective specific primary antibodie to detect them, and each band was also normalized to its corresponding beta-actin. We didn’t identify one indicator as multiple indicators. We could match the beta-actin to each indicator if needed.
19) Figure 6G: To show convincingly that PKM2 plays a role in the diabetic podocytes, double immunofluorescent staining should be performed for exact protein localization. The photomicrograph shows massive tubular PKM2 expression in db/db kidneys.
Response: Thank you for suggesting the addition of further experiments to verify thevalidity of our results. We have supplemented the results of immunofluorescence double staining experiments for PKM2 and the podocyte marker nestin. You can view the results in Fig. 6I on page 22 of the manuscript. We also added the corresponding text description to both Fig. 6I. We made revisions on pages 813 – 819 and lines 841 – 843 on page 22 of the manuscript.
20) PKM2 and m6A expression should be presented in the human kidney biopsies as well.
Response: Thank you very much for your guidance on our experiments. We have supplemented the results of immunofluorescence double staining experiments for PKM2 and the podocyte marker nestin. You can view the results in Fig. 6F on page 22 of the manuscript.
We also supplemented the results of the immunohistochemistry of m6A in the kidney tissue of DKD patients. You can view the results in Fig. 5I on page 20 of the manuscript. We also added corresponding written descriptions to both Fig. 5I and Fig. 6F. You can view Fig. 5I on pages 740-742 of page 20 and lines 771-772 of the manuscript. And see Fig. 6F in lines 800-809 on pages 21-22 and lines 834-839 on page 23 of the manuscript.
21) Demographic and basic clinical data of renal biopsy patietns should be added, including histology diagnosis, CKD staging , eGFR, proteinuria, medications.
Response: Thank you very much for your critical comments for us. We have added a detailed description of the patients on renal biopsy in lines 236 – 253 on page 6. I have supplemented the relevant results for DKD patients eGFR and UPR in the results section and described the results accordingly in the manuscript. You can view the results on page 14 of the manuscript in Figure 2C and Figure 2D and in the corresponding text description on page 13, lines 535-539 and lines 559-561 on page 15.
22) Study limitations need to be added before the conclusions.
Response: Thank you so much for pointing out our shortcomings. We have already described the limitations of the study in lines 1124-1132 on page 31 of the discussion section. The limitations of this study were as follows: First, although the mouse model provided valuable insights, extrapolation of the findings to human DKD patients may require further validation in clinical studies. Second, in conducting the investigation of molecular mechanisms, we performed and validated the mechanistic studies in vitro; however, we did not further validate them in gene-deficient mouse models, nor did we conduct any follow-up experiments in human cases of DKD.
23) A 38% match with other publications, accorring to plagiarism report (iThenticate) should be corrected, extensive revision of the draft is needed (please re-phrase those "copy/paste" sentences).
Response: Thank you for your constructive suggestion on our manuscript. We tried our best to improve the manuscript and made some changes to the manuscript. These changes will not influence the content and framework of the paper. And here we did not list the changes but marked in red in the revised paper.
Minor point: The many typos need to be corrected.
Response: Thank you for your valuable feedback on language improvements. We have completed for quick English editing provided by MDPl to help us modify spelling errors. I hope the paper can reach the standard of publication in language and we hope the revised manuscript could be acceptable for you. At the same time, we have added the English editing certificate in the attachment.
We sincerely appreciate the reviewers' attention and assistance in improving our research. Each suggestion is highly constructive and has provided us with direction for further improvement. We have responded to each comment in detail and made the corresponding modifications. Thank you once again for the assistance provided in improving our manuscript. We sincerely hope that these revisions meet your requirements and would be happy to engage in further discussions to ensure the quality of the paper. We hoping to meet your expectations. We look forward to any further feedback you may have. Thank you for your valuable support and time!
Reviewer 2 Report
Comments and Suggestions for Authors
This article investigates RNF183's role in diabetic kidney disease (DKD) through a multi-faceted approach. The authors propose a pathway where m6A RNA methylation regulates RNF183, which in turn ubiquitinates PKM2 for degradation. While the study provides extensive molecular evidence, several weaknesses in methodology and data presentation limit its impact.
1. The study uses db/db mice (a model of Type 2 diabetes) but does not specify whether human samples represent Type 1 or Type 2 diabetes. This inconsistency between the clinical and experimental models raises concerns about the applicability of the findings to human DKD. Additionally, the use of young (8-week-old) mice, which are unlikely to exhibit advanced kidney pathology, further limits the translational relevance.
2. Key markers of DKD progression, such as proteinuria, are not assessed in the animal studies. This incomplete characterization of kidney function impairs a comprehensive understanding of the disease context and hinders the evaluation of RNF183's role in DKD progression.
3. Section 4.1 (pages 522-523) contains errors in antibody documentation, including identical catalog numbers (sc-13551) for RNF183 and Nestin antibodies, which are from different vendors.
4. The description of the HPC (Human Podocyte Cells) experimental model lacks essential details, such as cell origin, establishment, characterization, and passage number. These omissions undermine the reliability and reproducibility of the podocyte-specific data.
5. The RNA sequencing analysis lacks comprehensive data presentation. While Figure 1A shows differential gene expression, the full dataset, including heatmaps and complete gene lists, is not provided. Additionally, the inclusion of RNF family phylogenetic analysis (1B) and organ distribution data (1D) appears unnecessary, as these bioinformatic analyses do not contribute to the study's core findings. Full RNA-seq data should be deposited in a public repository, and supplementary materials should include comprehensive gene expression data for improved transparency.
6. The proteomics analysis described in Section 2.6 is vague and unclear. It is not specified whether the authors exclusively analyzed Co-IP samples or also included protein extracts from RNF183 knockdown experiments. Furthermore, the absence of a complete list of identified proteins limits the exploration of pathways beyond PKM2 and RACK1. To enhance reproducibility and enable a thorough evaluation of RNF183's interactome, the full proteomics dataset should be made publicly available.
7. The study fails to cite a critical publication by Wu et al. (2018, PNAS, 115(12):E2762-E2771), which demonstrates RNF183's role in apoptosis through Bcl-xL degradation under ER stress. While the authors explore apoptosis, they do not measure Bcl-xL. This omission should be addressed by either measuring Bcl-xL levels or discussing how their findings align with or differ from this previous study.
8. Although PKM2 is known to support both metabolic reprogramming and stress responses, the authors’ data suggest it plays a detrimental role in this context. Citing studies by Wubben et al. (2020, Scientific Reports, 10(1):2990) and Bailleul et al. (2023, Neuro-Oncology, 25(11):1989-2000) to elaborate on PKM2's context-dependent roles could enhance the discussion and provide additional insights into its specific role in DKD.
9. The discussion section lacks critical analysis of the study’s limitations, particularly regarding the translational gap between animal models and human DKD. Differences in disease pathophysiology between humans and db/db mice could affect the applicability of the findings. The authors also fail to address important molecular-level questions raised earlier.
10. The statement "All data in this study are available from the corresponding author upon reasonable request" is suboptimal. Omics data should be deposited in a public repository to ensure accessibility and transparency.
Author Response
Reviewer #2:
We would like to express our sincere gratitude to the reviewers for their meticulous review and constructive feedback, which has given us the opportunity to improve our manuscript in various aspects. We have carefully addressed each comment and have made modifications accordingly. And here we have marked in red in the revised paper.
Q1: The study uses db/db mice (a model of Type 2 diabetes) but does not specify whether human samples represent Type 1 or Type 2 diabetes. This inconsistency between the clinical and experimental models raises concerns about the applicability of the findings to human DKD. Additionally, the use of young (8-week-old) mice, which are unlikely to exhibit advanced kidney pathology, further limits the translational relevance.
Response: Thank you for your constructive suggestion. The kidney biopsy specimens we collected are from middle-aged and elderly individuals, all of whom have type 2 diabetes. We have provided a detailed description of this in the Materials and Methods section. Although we purchased the mice at eight weeks of age, we kept them until sixteen weeks before harvesting tissues and euthanizing them. We have also made a detailed update regarding this in the Materials and Methods section. You can view our changes at line 269, page 6 of the manuscript.
Q2: Key markers of DKD progression, such as proteinuria, are not assessed in the animal studies. This incomplete characterization of kidney function impairs a comprehensive understanding of the disease context and hinders the evaluation of RNF183's role in DKD progression.
Response: Thank you for suggesting the addition of further experiments to verify thevalidity of our results. We collected urine from diabetic mice for 24 h in metabolic cages and performed 24 h albuminuria and urinary creatinine. Reagents used for the urine testing have been supplemented in Materials Method 2.1. The corresponding results are added in the Fig. 1D, and the corresponding data are described in the Results. You can view it in the Results section page 12, lines 511-514 and 523-524.
Q3: Section 4.1 (pages 522-523) contains errors in antibody documentation, including identical catalog numbers (sc-13551) for RNF183 and Nestin antibodies, which are from different vendors.
Response: Firstly I am sorry for the negligence and mistake. I have corrected the catalog number for the RNF183 (ARP43404) and nestin (sc-23927) antibodies. You can check it out on page 4 of the manuscript, lines 182-186.
Q4: The description of the HPC (Human Podocyte Cells) experimental model lacks essential details, such as cell origin, establishment, characterization, and passage number. These omissions undermine the reliability and reproducibility of the podocyte-specific data.
Response: Thanks for your friendly suggestions for our work. We have added a more detailed explanation to the manuscript. We have supplemented therelevant sections based on your feedback, aiming to provide readers with amore comprehensive understanding of our methods. The supplementation of these data will enhance the reliability and reproducibility of our study. Details can be viewed in lines 307-327 on pages 7-8 of the manuscript.
Q5: The RNA sequencing analysis lacks comprehensive data presentation. While Figure 1A shows differential gene expression, the full dataset, including heatmaps and complete gene lists, is not provided. Additionally, the inclusion of RNF family phylogenetic analysis (1B) and organ distribution data (1D) appears unnecessary, as these bioinformatic analyses do not contribute to the study's core findings. Full RNA-seq data should be deposited in a public repository, and supplementary materials should include comprehensive gene expression data for improved transparency.
Response: Thanks for your suggestions in the results section. We have carefully adopted your advice and have made modifications accordingly. We have uploaded the complete gene list of Fig. 1A to Supplementary Material 1, and we have removed Fig. 1B and Fig. 1C. We will store the complete RNA-seq data in a public repository immediately after the article is received by the journal.
Q6: The proteomics analysis described in Section 2.6 is vague and unclear. It is not specified whether the authors exclusively analyzed Co-IP samples or also included protein extracts from RNF183 knockdown experiments. Furthermore, the absence of a complete list of identified proteins limits the exploration of pathways beyond PKM2 and RACK1. To enhance reproducibility and enable a thorough evaluation of RNF183's interactome, the full proteomics dataset should be made publicly available.
Response: Sorry to make the inconvenience and I have re-described the proteomics analysis in detail in Section 3.6 of the manuscript. Meanwhile, we have uploaded the full gene list of Fig. 6A and Fig. 6B to Supplementary Material 2 and 3. From these analyses, we first downregulated RNF183 under normal glucose conditions and performed LC-MS analysis on the whole cell protein extracts. Among the differentially expressed proteins, we selected 343 proteins that were upregulated in the RNF183 knockdown group (Fig. 6A). At the same time, after upregulating RNF183 under normal glucose conditions, we conducted Co-IP experiments combined with LC-MS analysis and selected 23 proteins that directly bound to RNF183 (Fig. 6B). You can also view this detail on page 21, lines 776-788. After the journal receives the article, we will immediately store the complete proteomics dataset in a public repository.
Q7: The study fails to cite a critical publication by Wu et al. (2018, PNAS, 115(12):E2762-E2771), which demonstrates RNF183's role in apoptosis through Bcl-xL degradation under ER stress. While the authors explore apoptosis, they do not measure Bcl-xL. This omission should be addressed by either measuring Bcl-xL levels or discussing how their findings align with or differ from this previous study.
Response: Thank you very much for your kind guidance. First, cell lines and disease models are not the same, and RNF183 may mediate different apoptosis. Secondly, Bcl-2 is a ubiquitously expressed protein that also has anti-apoptotic effects. For Bcl-2 and Bcl-xL to inhibit apoptosis by different mechanisms, given that Bcl-2 and Bcl-xL have similar anti-apoptotic functions, we represented the level of apoptosis by examining BCL-2 and BAX.
Q8: Although PKM2 is known to support both metabolic reprogramming and stress responses, the authors’ data suggest it plays a detrimental role in this context. Citing studies by Wubben et al. (2020, Scientific Reports, 10(1):2990) and Bailleul et al. (2023, Neuro-Oncology, 25(11):1989-2000) to elaborate on PKM2's context-dependent roles could enhance the discussion and provide additional insights into its specific role in DKD.
Response: Thank you for your constructive suggestion on our manuscript. I have accepted your suggestion and quoted these two articles in the Discussion section. You can view the detailed corrections on page 30, lines 1065-1073 in the manuscript.
Q9: The discussion section lacks critical analysis of the study’s limitations, particularly regarding the translational gap between animal models and human DKD. Differences in disease pathophysiology between humans and db/db mice could affect the applicability of the findings. The authors also fail to address important molecular-level questions raised earlier.
Response: Thank you so much for pointing out our shortcomings. We have already described the limitations of the study in lines 1124-1132 on page 31 of the discussion section. The limitations of this study were as follows: First, although the mouse model provided valuable insights, extrapolation of the findings to human DKD patients may require further validation in clinical studies. Second, in conducting the investigation of molecular mechanisms, we performed and validated the mechanistic studies in vitro; however, we did not further validate them in gene-deficient mouse models, nor did we conduct any follow-up experiments in human cases of DKD.
Q10: The statement "All data in this study are available from the corresponding author upon reasonable request" is suboptimal. Omics data should be deposited in a public repository to ensure accessibility and transparency.
Response: Thank you very much for your suggestions and guidance. After the journal receives the article, we will immediately store the complete RNA-seq data and the complete proteomics dataset in a public repository.
We sincerely appreciate the reviewers' attention and assistance in improving our research. Each suggestion is highly constructive and has provided us with direction for further improvement. We have responded to each comment in detail and made the corresponding modifications. Thank you once again for the assistance provided in improving our manuscript. We sincerely hope that these revisions meet your requirements and would be happy to engage in further discussions to ensure the quality of the paper. We hoping to meet your expectations. We look forward to any further feedback you may have. Thank you for your valuable support and time!
Reviewer 3 Report
Comments and Suggestions for Authors
Dear all,
The comments follow throughout the attached document.

Author Response
Reviewer #3:
We would like to express our sincere gratitude to the reviewers for their meticulous review and constructive feedback, which has given us the opportunity to improve our manuscript in various aspects. Each suggestion is highly constructive and has provided us with direction for further improvement. We have responded to each comment in detail and made the corresponding modifications. And here we have marked in red in the revised paper. Thank you once again for the assistance provided in improving our manuscript. We sincerely hope that these revisions meet your requirements and would be happy to engage in further discussions to ensure the quality of the paper. We hoping to meet your expectations. We look forward to any further feedback you may have. Thank you for your valuable support and time!
Round 2
Reviewer 1 Report
Comments and Suggestions for Authors
The authors have revised the manuscript and have significantly improved several crucial chapters. However, data presentation have been improved only in some figures, and this needs to be revised.
The simple bar-charts have not been changed to scatter-plots to show individual experimental data in following Figures: 3C, 3E (LC3, p62 and BCL western blots), 4C, 4F, 4I, 5A, 5D-5G (in all grey charts of western blots), 6E (all grey charts), 7C-7E (grey charts).
Figure 8E should be separatedly shown as individual figure as proposed pathomechanism, and the figure legend should include a clear description. I suggest to place that explanatory figure along with the study conclusions in the discussion chapter.
The plagiarism report of the revised draft shows a 6% reduction but still quite high (32%), for instance protein extraction method is almost 100% identically found in a paper of Fang Li et al. (https://doi.org/10.1016/j.biocel.2021.106117). You may try to re-phrase those sentences taken from other publication.
Author Response
Reviewer #1:
We greatly appreciate for your valuable comments. We have carefully considered all comments from the reviewer and revised our manuscript accordingly. we have marked in red in the revised paper.
- The simple bar-charts have not been changed to scatter-plots to show individual experimental data in following Figures: 3C, 3E (LC3, p62 and BCL western blots), 4C, 4F, 4I, 5A, 5D-5G (in all grey charts of western blots), 6E (all grey charts), 7C-7E (grey charts).
Response: Thank you for your insightful comment and kind suggestion. We have changed all the bar charts in the results to scatter bar charts. You can view these modifications in our results (Fig. 3, Fig. 4, Fig. 5, Fig. 6, Fig. 7). You can see the modified results on page 16, 18, 20, 22 and 25 .
2) Figure 8E should be separatedly shown as individual figure as proposed pathomechanism, and the figure legend should include a clear description. I suggest to place that explanatory figure along with the study conclusions in the discussion chapter.
Response: Thanks for your friendly suggestions for our work. We have changed Figure 8E to Figure 9. The modifications are as follows: In HPC cells stimulated with high glucose, high glucose induced a decrease in METTL3 and IGF2BP2 levels, which mediated changes in the transcriptional levels of RNF183, leading to reduced RNF183 mRNA expression. Downregulated expression of RNF183 under high glucose conditions, leading to increased protein levels of PKM2, decreases autophagy and promotes apoptosis. You can view these modifications at lines 1128-1134, page 31 of the manuscript.
3) The plagiarism report of the revised draft shows a 6% reduction but still quite high (32%), for instance protein extraction method is almost 100% identically found in a paper of Fang Li et al. (https://doi.org/10.1016/j.biocel.2021.106117). You may try to re-phrase those sentences taken from other publication.
Response: Thank you for your constructive suggestion on our manuscript. We feel very guilty about this issue, and we have done our best to revise all of the manuscript. We have referred to a lot of literature to revise the high repetition rate of the manuscript. The corrections were marked in red and the repeat rate was detected using software. We sincerely hope that these revisions meet your requirements.
We sincerely thank the reviewers for their suggestions and assistance with our study. Each suggestion is highly constructive and we have made the corresponding modifications as recommended. Thank you once again for the assistance provided in improving our manuscript. We sincerely hope that these revisions meet your requirements and would be happy to engage in further discussions to ensure the quality of the paper. We hoping to meet your expectations. We look forward to any further feedback you may have. Thank you for your valuable support and time!

Reviewer 2 Report
Comments and Suggestions for Authors
The authors have clarified using Type 2 diabetic patients, extended the mouse model to 16 weeks, added critical albuminuria data, corrected antibody documentation, and improved data presentation with supplementary materials. Overall, the manuscript quality has been significantly enhanced.
One concern remains: the source information (HPC,CE28120) of HPC cells provided doesn't appear in public databases, making it unclear whether these are immortalized cell lines or primary cultures from a service provider. We recommend specifying whether the cells were commercially obtained as established lines or custom-cultured from tissue samples.
Author Response
Reviewer #2:
We greatly appreciate for your valuable comments. We have carefully considered all comments from the reviewer and revised our manuscript accordingly. we have marked in red in the revised paper.
1) One concern remains: the source information (HPC,CE28120) of HPC cells provided doesn't appear in public databases, making it unclear whether these are immortalized cell lines or primary cultures from a service provider. We recommend specifying whether the cells were commercially obtained as established lines or custom-cultured from tissue samples.
Response: Thank you for suggesting that we add more detailed explanations in the 2.5. Cell culture and groups. We have supplemented the source information of HPC cells based on your feedback, aiming to provide readers with a more comprehensive understanding of our methods. The modifications are as follows: Human podocytes (HPC, CE28120) were purchased from Beijing Keruisi Bio-Tech Co., Ltd., China, and the cells were maintained in the laboratory. The immortalized HPC cells were achieved by introducing the hTERT gene. hTERT is a gene that encodes the catalytic subunit of the telomerase protein, which is positively correlated with telomerase activity. By viral-mediated expression of hTERT, telomerase in normal cells was activated, leading to the elongation of telomeric DNA and the extension of the cell cycle, thus achieving cell immortalization. HPC cells were cultured in RPMI-1640 at 37°C with 5% COâ‚‚, supplemented with 10% fetal bovine serum, 100 U/mL penicillin, and 100 μg/mL streptomycin. HPC cells were adherent cells that grew in a typical (epithelial) cobblestone morphology. The medium was changed every 2-3 days, and cells were passaged every 3-4 days at a ratio of 1:2. This content can also be viewed by you in lines 313-327 on pages 7-8 of the manuscript.
We sincerely thank the reviewers for their suggestions and assistance with our study. Each suggestion is highly constructive and we have made the corresponding modifications as recommended. Thank you once again for the assistance provided in improving our manuscript. We sincerely hope that these revisions meet your requirements and would be happy to engage in further discussions to ensure the quality of the paper. We hoping to meet your expectations. We look forward to any further feedback you may have. Thank you for your valuable support and time!

Reviewer 3 Report
Comments and Suggestions for Authors
Suggestions:
In line 272, please indicate which euthanasia technique was used.
Put in italics:
Line 569, 979
Put the acronym CKD, in line 989.
Author Response
Reviewer #3:
We greatly appreciate for your valuable comments. We have carefully considered all comments from the reviewer and revised our manuscript accordingly. we have marked in red in the revised paper.
- In line 272, please indicate which euthanasia technique was used.
Response: Thank you for suggesting that we add more detailed explanations in the 2.3. Animals. We have supplemented the relevant information about mouse euthanasia based on your feedback, aiming to provide readers with a more comprehensive understanding of our methods. The modifications are as follows: All mice were anesthetized with isoflurane, and experiments were conducted once the mice lost consciousness. After the experiment, the experimenter performed cervical dislocation on the mice. The thumb and index finger were placed on both sides of the base of the skull at the neck of the mouse, and with the other hand, the tail was quickly pulled at the base, resulting in the separation of the cervical vertebrae from the skull for euthanasia. This content can also be viewed by you in lines 272-279 on pages 6-7 of the manuscript.
2) Put in italics:
Line 569, 979
Response: Firstly I am sorry for the negligence and mistake. I have changed them to italics. You can check these corrections on page 15, lines 581 and on page 28, lines 984 of the manuscript.
3) Put the acronym CKD, in line 989.
Response: We sincerely thank you for your careful reading and for pointing out our mistakes. We have changed the chronic kidney disease in the manuscript to the acronym CKD. You can view it on page 28, line 1002 of the manuscript.
We sincerely thank the reviewers for their suggestions and assistance with our study. Each suggestion is highly constructive and we have made the corresponding modifications as recommended. Thank you once again for the assistance provided in improving our manuscript. We sincerely hope that these revisions meet your requirements and would be happy to engage in further discussions to ensure the quality of the paper. We hoping to meet your expectations. We look forward to any further feedback you may have. Thank you for your valuable support and time!

Round 3
Reviewer 1 Report
Comments and Suggestions for Authors
The authors successfully revised the draft as suggested.